# ALPHA DISCOVERY VIA GRAMMAR-GUIDED LEARNING AND SEARCH

## ABSTRACT

Automatically discovering formulaic alpha factors is a central problem in quantitative finance. Existing methods often ignore syntactic and semantic constraints, relying on exhaustive search over an unbounded and unstructured space that limits performance and interpretability. We present AlphaCFG, the first framework for defining and discovering alpha factors that are syntactically valid, financially interpretable, and computationally efficient. In this framework, we first define an alpha-oriented Context-Free Grammar (CFG) to construct a tree-structured, size-controlled search space of human-interpretable alpha expressions, enabling grammar-tailored search and learning. We then formulate the search of high-performance alphas in this space as a very large, tree-structured linguistic Markov Decision Process (TSL-MDP), where each leaf state is an alpha expression with its information coefficient as reward. To efficiently navigate the TSL-MDP, we develop syntax-similarity-based representation learning to estimate alpha expression performance (value network) and grammar production rule probabilities (policy network), and integrate it into a grammar-aware Monte Carlo Tree Search. Experiments on China and US stock markets' datasets show that AlphaCFG outperforms state-of-the-art baselines in both search efficiency and trading profitability. AlphaCFG also provides an easy-to-use approach for refining and improving existing formulaic alpha factors.

## 1 INTRODUCTION

### 1.1 ALPHA DISCOVERY

In quantitative finance, alpha factors are critical for addressing several key challenges, particularly in asset management, quantitative trading, and investment strategy development. They are functions that map the features (e.g., trading volume, highest price, lowest price, etc.) of a stock over a period of trading days to a prediction of its future return. Alpha discovery is the systematic process of identifying new functions that can predict investment returns based on historical data.

Alpha discovery methods can be broadly classified into three categories. First, *heuristic or expert-driven methods* were mainly based on domain knowledge, such as value factors (e.g., price-to-earnings Ratio (Fama & French, 1992)) and momentum factors (e.g., total return of past 12 months (Carhart, 1997)). While these handcrafted alpha factors help extract expected return signals, they rely on limited heuristics and lack a sustainable discovery framework. Moreover, widespread use of these simple alphas in the investment market leads to rapid arbitrage, reducing predictive accuracy over time. Second, *data-driven learning methods* include statistical approaches (e.g., regression (Panwar et al., 2021)), supervised learning (e.g., tree-based ensembles (Almaafi et al., 2023)), as well as unsupervised learning (Babu et al., 2012) and reinforcement learning (Lee, 2001). These methods enable the discovery of complex, nonlinear patterns in financial data. However, a key challenge is their black-box nature, which often leads to poor explainability and an increased risk of overfitting in the discovered alphas. Third, *formulaic alpha methods* (Kakushadze, 2016) emphasize human-readable alphas. Formulaic alpha factors are explicit mathematical expressions that map raw financial inputs—such as price and volume—into scalar values. These formulaic expressions are typically composed using a predefined set of operators and functions (e.g., ranks, differences, moving averages). While the concept is not new, it has recently regained attention due to its potential to yield interpretable and transparent alphas.

Our work lies at the intersection of the second and third categories, aiming at the *automatic discovery of explainable alphas*. This task can be seen as *symbolic regression* (Makke & Chawla, 2024), which aims at discovering explicit mathematical expressions that optimally fit the data, overcoming the uninterpretability of black-box models. Early approaches such as genetic programming (GP) (Zhang et al., 2020) optimize the information coefficient by evolving expression trees, but suffer from exponential search growth and local optima. More recently, AlphaGen (Yu et al., 2023) applies reinforcement learning to iteratively generate factors and combine them into composite pools, while AlphaQCM (Zhu & Zhu, 2025) extends this idea with distributed RL to improve scalability.

Existing methods for the automatic discovery of formulaic alphas face fundamental challenges.

(1) Automated discovery of formulaic alphas essentially involves searching for mathematical languages, where a linguistic framework could enhance the search. However, such a framework is lacking in the literature. Without formal linguistic guidance, current methods must exhaustively explore a vast combinatorial, even infinite space of sequences, relying on informal syntactic checks for factor validity. This results in *limited accuracy, low performance and computation inefficiency*.

(2) Different mathematical sequences can represent semantically equivalent expressions, yet current methods use linear networks to encode sequence that redundantly includes such variants. Consequently, *existing methods spend effort on seemingly distinct sequences that encode the same meaning, greatly reducing efficiency*.

## 1.2 OUR WORK

We propose AlphaCFG, the first grammar–based framework for automated alpha discovery. By combining Context-Free Grammar (CFG) (Chomsky & Schützenberger, 1963) with Monte Carlo Tree Search (MCTS) (Chaslot, 2010) under syntax-aware representation learning, AlphaCFG provides a principled system for generating, validating, and interpreting high-performance alphas.

(1) *Grammar-Constrained Alpha Factors*. We introduce $\alpha$-CFG-Sem-$k$, a formal language that integrates CFG with domain knowledge of alphas. It recursively generates expressions that are structurally valid and financially meaningful, with two built-in mechanisms: (i) length constraints to bound the search space, and (ii) expression-tree pruning to eliminate syntactically different but semantically equivalent factors. This resolves core difficulties of alpha mining—invalid structures, semantic redundancy, and unbounded exploration.

(2) *Structured Characterization of Alpha Space*. Based on $\alpha$-CFG-Sem-$k$, we formulate alpha discovery as a Tree-Structured Linguistic MDP (TSL-MDP), where each leaf state is a candidate expression and rewards are defined by information coefficient (IC). TSL-MDP provides a characterization of the grammar-guided, interpretable, and scalable search space, that enables efficient search and learning algorithm design.

(3) *Reinforcing MCTS with Syntax-Representation Learning*. To solve the above TSL-MDP, we design a grammar-aware MCTS augmented with structure-aware neural representations. A grammar-guided Upper Confidence Bound algorithm (Auer et al., 2002) drives edge selection, while a Tree-LSTM (Tai et al., 2015) encodes each state into features shared by two networks: a value network that learns from trading data to evaluate states, and a policy network that guides searching. MCTS iteratively updates with these evaluations, yielding stronger policies and more effective alpha discovery.

The objective of AlphaCFG is to establish a general and flexible "linguistic theory + machine learning" framework for generating formulaic alpha factors. It is not restricted to high-performing trading strategy. It can be applied to other tasks such as risk modelling, portfolio construction, and asset pricing. Users can utilize their domain knowledge to set the operators and loss functions in AlphaCFG. However, to showcase the advantage of AlphaCFG, we empirically evaluate it trading performance on CSI 300 and S&P 500 stocks. Using returns, IC, Sharpe ratio, and maximum drawdown, we confirm the superior profitability of the discovered factors via AlphaCFG. Detailed results also show that refinement of CFG yields faster convergence and higher-quality factors. We also conduct separate ablation studies to verify the importance of grammar design for factor generation and the effectiveness of syntax-representation learning. Moreover, starting from partial states, our method effectively strengthens predictive performance of existing factors.

## 2 PROBLEM FORMULATION

Consider a market with $n$ stocks over $T$ trading days. For each day $t \in \{1, 2, \ldots, T\}$, stock $i$ has a feature matrix $\mathbf{x}_{t,i} \in \mathbb{R}^{m \times \tau'}$, consisting of $m$ raw features (e.g., opening/closing prices) over the current and previous $\tau' - 1$ days. An alpha factor $f$ maps the feature tensor $\mathbf{X} = [\mathbf{x}_{t,1}, \mathbf{x}_{t,2}, \ldots, \mathbf{x}_{t,n}] \in \mathbb{R}^{n \times m \times \tau'}$ to a vector $\mathbf{y} = f(\mathbf{X}) \in \mathbb{R}^n$ (shown in Figure 1a). The alpha value for stock $i$ on day $t$ is $y_{t,i} = f(\mathbf{x}_{t,i})$. Formulaic factors (shown in Figure 1b) are just factors constructed by operators (Table 5) along with predefined constants (Table 4) and features (Table 3). These symbols come from a set of operators and operands (Yang et al., 2020) commonly used in the field of formulaic factors.

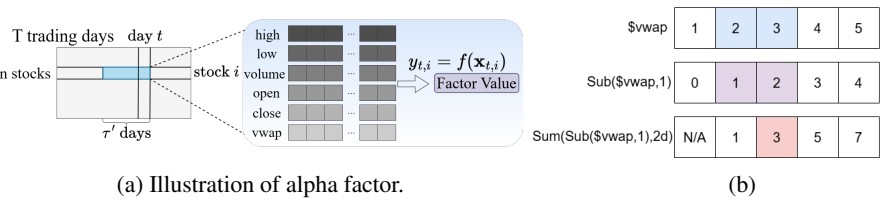

(a) Illustration of alpha factor.            (b)

Figure 1: (a) Illustration of alpha factor. (b) An example of formulaic factor: The factor $\mathrm{Sum}(\mathrm{Sub}(vwap, 1), 2d)$ computes the sum of the most recent two days of VWAP values after subtracting 1 from each. To obtain the factor value on Wednesday, the operator first evaluates $\mathrm{Sub}(vwap, 1)$ for Tuesday and Wednesday and then aggregates them: $(2 - 1) + (3 - 1) = 3$. This output serves as the alpha signal, the predicted return for Wednesday which is subsequently used in downstream stock-selection or portfolio-construction procedures.

The primary objective of an alpha factor $f$ is to predict future stock returns. The standard metric for assessing factor quality is IC, defined as the cross-sectional correlation between factor values and subsequent realized returns (Grinold & Kahn, 2000). The $\tau$-day realized return of stock $i$ observed on day $t$ is $r_{t,i}^{(\tau)} = \frac{\mathrm{Close}_{t+\tau,i}}{\mathrm{Close}_{t,i}} - 1$, where $\mathrm{Close}_{t,i}$ denotes the closing price of stock $i$ on day $t$. Let the cross-sectional factor vector and the realized return vector of $n$ stocks on day $t$ be $\mathbf{y}_t = (y_{t,1}, \ldots, y_{t,n})$ and $\mathbf{r}_t^{(\tau)} = (r_{t,1}^{(\tau)}, \ldots, r_{t,n}^{(\tau)})$, respectively. Then, the daily IC is the Pearson correlation coefficient between the factor values and the realized returns:

$$\mathrm{IC}_t(\mathbf{y}_t, \mathbf{r}_t^{(\tau)}) = \frac{\sum_{i=1}^n (y_{t,i} - \bar{y}_t)(r_{t,i}^{(\tau)} - \bar{r}_t^{(\tau)})}{\sqrt{\sum_{i=1}^n (y_{t,i} - \bar{y}_t)^2} \sqrt{\sum_{i=1}^n (r_{t,i}^{(\tau)} - \bar{r}_t^{(\tau)})^2}}, \tag{1}$$

where $\bar{y}_t = \frac{1}{n} \sum_{i=1}^n y_{t,i}$ and $\bar{r}_t^{(\tau)} = \frac{1}{n} \sum_{i=1}^n r_{t,i}^{(\tau)}$.

To evaluate the performance of factor $f$ over $T$ days, we consider $\mathrm{IC}(f) = \frac{1}{T} \sum_{t=1}^T \mathrm{IC}_t(\mathbf{y}_t, \mathbf{r}_t^{(\tau)})$, the average daily IC as the IC of factor $f$, where higher $\mathrm{IC}(f)$ indicates stronger predictive power of factor $f$. *Then, the task of alpha discovery is to find a factor with as high an IC as possible.*

It is worth mentioning that finding a single high-IC formulaic factor is difficult, but combining multiple factors linearly is more effective: it eases the search, improves prediction, and preserves interpretability. Following Alphagen (Yu et al., 2023), we optimize the IC of such linear combinations (the *factor pool*) by this approach (see Algorithm 1 in Appendix B.1).

## 3 LANGUAGE CHARACTERIZATION OF INTERPRETABLE ALPHAS

The number of potential formulaic alphas grows combinatorially with expression length, making brute-force search highly inefficient. Moreover, many alpha candidates are either *syntactically invalid* or *semantically nonsensical*, hindering effective and interpretable alpha discovery. To address these challenges, we use *Context-Free Grammar (CFG)* (Hopcroft & Ullman, 1979) to formally characterize the alpha-factor search space.

**Definition 1** (CFG). A context-free grammar $G$ is a tuple $G = (\mathcal{N}, \mathcal{T}, \mathcal{P}, \mathcal{S})$, where $\mathcal{N}$ is a finite set of *nonterminal symbols*; $\mathcal{T}$ is a finite set of *terminal symbols*, with $\mathcal{N} \cap \mathcal{T} = \emptyset$; $\mathcal{P} \subseteq \mathcal{N} \times (\mathcal{N} \cup \mathcal{T})^*$

is a set of *production rules*, each written in the form $\Gamma \to \beta$ where $\Gamma \in \mathcal{N}, \beta \in (\mathcal{N} \cup \mathcal{T})^*$; $\mathcal{S} \in \mathcal{N}$ is the *start symbol*, from which the derivation of expressions begins.

To construct a formula, from the start symbol, a CFG recursively applies the production rules $\Gamma \to \beta$ to replace the leftmost nonterminal symbol $\Gamma$ with a sequence $\beta \in (\mathcal{N} \cup \mathcal{T})^*$ (*leftmost derivation* (Hopcroft & Ullman, 1979)), until only terminal symbols remain. Unlike *Reverse Polish Notation (RPN)* (Krtolica & Stanimirović, 2004), CFG (1) guarantees syntactic validity, (2) supports semantic constraints, (3) enables explicit control of expression length via derivation depth, and (4) provides a hierarchical structure mapping to abstract syntax trees. These ensure interpretability and make efficient search possible.

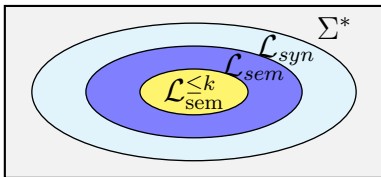

Figure 2: Different spaces for alphas. $\Sigma^*$: expressions with all possible combinations of symbols; $\mathcal{L}_{\mathrm{syn}}$: syntactically valid alphas; $\mathcal{L}_{\mathrm{sem}}$: semantically valid alphas; $\mathcal{L}_{\mathrm{sem}}^{\leq K}$: semantically valid length $\leq K$ alphas.

### 3.1 Syntactically Valid Alpha Generator

To achieve syntactic validity, we require generated alpha expressions to satisfy two conditions: (i) the structure is well-formed, enforced by a prefix notation and a recursive nonterminal-expansion scheme; and (ii) the operator-operand arity is consistent, ensuring that each operator receives exactly the required number of operands. Thus we adopt the following form:

$$\mathsf{Expr} \to \mathsf{Op}(\mathsf{Expr}, \dots) \mid \mathsf{TermSyb}, \tag{2}$$

where $\mathsf{Expr} \in \mathcal{N}$ denotes a recursively constructible nonterminal class of expressions, $\mathsf{Op} \in \mathcal{T}$ denotes the prefix-notaion operators, and $\mathsf{TermSyb} \in \mathcal{T}$ denotes the features and constants.

**Structure**. In terms of the structure, Formula (2) enforces that all alpha factors must adopt the *prefix-notation* style, where each operator $\mathsf{Op}$ precedes its operands (i.e., symbols in the parenthesis). This formation allows $\mathsf{Expr}$ to be *recursively* expanded through nested applications of $\mathsf{Op}$ (i.e., $\mathsf{Expr}$ inside the parentheses), while also permitting *termination* of the recursive process by substituting features or constants (i.e., terminal symbols that do not lead to further nesting), thereby completing the construction of the expression. Taken together, prefix notation, recursive expansion, and termination eliminate any ambiguity in the order of operations, allow complex and informative alpha expressions to be constructed from *a small set of primitives*, naturally map the alpha expressions to *tree structures*, which allow further tree-based search algorithms and machine learning methods.

**Arity**. As alpha factors are constructed in quantitative trading setting, we instantiate $\mathsf{Op}$ by operator families with fixed arity, including unary ($\mathsf{UnaryOp}$), binary operators ($\mathsf{BinaryOp}$), rolling operators ($\mathsf{RollingOp}$), paired rolling ($\mathsf{PairedRollingOp}$), and nullary operators with zero operand for constants and features ($\mathsf{TermSyb}$). The corresponding production patterns are as Formula (3):

$$\mathsf{Expr} \to \mathsf{UnaryOp}(\mathsf{Expr}) \mid \mathsf{BinaryOp}(\mathsf{Expr}, \mathsf{Expr}) \mid \mathsf{RollingOp}(\mathsf{Expr}, \mathsf{Expr})$$
$$\mid \mathsf{PairedRollingOp}(\mathsf{Expr}, \mathsf{Expr}, \mathsf{Expr}) \mid \mathsf{TermSyb} \tag{3}$$

In the Appendix, Table 5 enumerates all operator symbols used in alpha factors together with their corresponding arity categories. Table 4 and Table 3 list all constants and features. Building on the structural and arity rules introduced above, we now provide the formal definition of $\alpha$-CFG-Syn.

**Definition 2** ($\alpha$-CFG-Syn). The context-free grammar for alpha factor expressions is a tuple $G = (\mathcal{N}, \mathcal{T}, \mathcal{P}, \mathcal{S})$, where: $\mathcal{N}$ is a set of nonterminal symbols; $\mathcal{T}$ is a set of *terminal symbols*, which includes: all operators listed in Table 5, and a fixed set of constants listed in Table 4, and a set of features listed in Table 3; $\mathcal{P}$ is a set of *production rules* in the forms illustrated in Formula (3), where the 'XxOp' symbols are replaced with specific symbols in Table 5; $\mathcal{S} \in \mathcal{N}$ is the start symbol, a uniquely designated nonterminal symbol $\mathsf{Expr}$ from which the derivation of strings begins.

## 3.2 FINANCE SEMANTICALLY-INTERPRETABLE ALPHA GENERATOR

While $\alpha$-CFG-Syn guarantees syntactic validity, it does not ensure semantic soundness in quantitative trading. Many syntactically valid expressions still violate financial logic. We therefore extend Definition 2 with domain-informed semantic constraints: (1) *Rolling Window*: the last operand of RollingOp and PairedRollingOp must be an integer constant (fixed window size). (2) *Constant Nesting*: pure constant–operator expressions (e.g., Add(0.1, 0.2)) are excluded as trivial. (3) *Numerical Stability*: operators like Log require domain-restricted inputs to avoid undefined values. (4) *Rolling Operand*: PairedRollingOp must take two time-series features; constants are disallowed since they lack variation. To encode these rules, we introduce three nonterminals: Num for rolling window sizes, Constant for numerical values, and Feature for stock-derived variables.

**Definition 3** ($\alpha$-CFG-Sem). The context-free grammar for generating semantic alpha factor expressions is defined as $G = (\mathcal{N}, \mathcal{T}, \mathcal{P}, \mathcal{S})$.

1. $\mathcal{N} = \{\text{Expr}, \text{Constant}, \text{Num}\}$ is the set of *nonterminal symbols*.

2. $\mathcal{T}$ is the set of *terminal symbols*, containing all operators (see Table 5), all features (see Table 3), and all the predefined constant (see Table 4).

3. $\mathcal{P}$ is the set of production rules that distinguishes the type of operands. (The productions use type operators as placeholders for specific operators to illustrate their production rules. The mapping between specific types and operators is shown in Table 5. [1])

$$
\begin{aligned}
\text{Expr} \ \rightarrow \ &\text{Feature} \mid \text{UnaryOp}(\text{Expr}) \\
&\mid \text{BinaryOp}(\text{Expr}, \text{Expr}) \mid \text{BinaryOp}(\text{Expr}, \text{Constant}) \mid \text{BinaryOp\_Asym}(\text{Constant}, \text{Expr}) \\
&\mid \text{RollingOp}(\text{Expr}, \text{Num}) \mid \text{PairedRollingOp}(\text{Expr}, \text{Expr}, \text{Num}) \\
\text{Num} \ \rightarrow \ &20 \mid \ldots \qquad \text{Constant} \ \rightarrow \ -0.01 \mid \ldots
\end{aligned}
$$

4. $\mathcal{S} = \text{Expr}$ is the start symbol, from which the derivation begins.

Although $\alpha$-CFG-Sem enforces syntactic and semantic validity, its recursive rules may still produce unbounded expressions and an intractable search space. To control this, we introduce a *k-bounded constraint* (Jin et al., 2018), which maintains a counter $k$ and caps it at $K$. Each production rule contributes an increment $\Delta k$ to the length of the expression (see Table 6), and a rule is applied only if $k+\Delta k \leq K$. This guarantees bounded expansions, yielding grammar $\alpha$-CFG-Sem-K (Algorithm 2).

## 3.3 CHARACTERIZING THE SPACE STRUCTURE OF $\alpha$-CFG-SEM-$k$

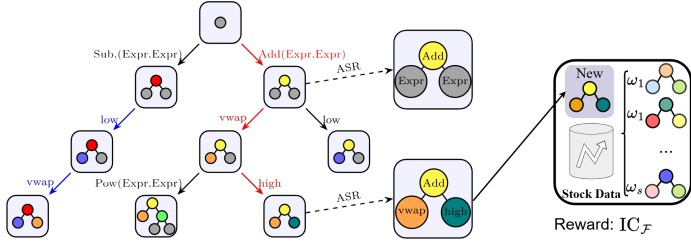

Figure 3: The alpha search space is as a huge tree. ASR is the zoomed round-box.

The set of all grammar-generated alphas forms a formal *alpha language*. The CFG-based syntactic, semantic, and length-bounded variants of languages, i.e., $\mathcal{L}_{\text{syn}}$, $\mathcal{L}_{\text{sem}}$, and $\mathcal{L}_{\text{sem}}^{\leq K}$, are nested as shown in Figure 2. Each language layer defines a progressively reduced search space for alpha factors. $\alpha$-CFG-Sem-$k$ makes it possible to explicitly characterize the structure of these search spaces. We provide a more rigorous complexity analysis in Appendix D.

---

[1]Symbols in Table 5, Table 3, Table 4 and Table 5 are not limited and can be extended by adding any additional operators, features, or constants relevant to the specific domain or task.

**Definition 4** (Search Space Structure). Given a grammar $\alpha$-CFG-Syn, $\alpha$-CFG-Sem, or $\alpha$-CFG-Sem-$k$, the search space of all possible alpha factors can be represented as a large tree: the root is the start symbol; each edge is a production step; intermediate nodes denote partially derived expressions; and leaf nodes are fully derived alpha factors.

Alpha discovery requires searching formulas in the infinite space $\Sigma^*$, whose unstructured nature makes efficient exploration infeasible. We reformulate it as the preorder language $\mathcal{L}_{\text{sem}}^{\leq k}$ corresponding to $\alpha$-CFG-Sem-$k$, which forms a natural tree-structured space—*reducing discovery to finding high-quality nodes within a large tree*. Figure 3 illustrates this space: each round-box node is an expression with an Abstract Syntax Representation (ASR)[2], where grey nodes are nonterminals, colored nodes are terminals, and edges denote operations. Expansions via production rules capture the recursive process of $\alpha$-CFG-Sem-$k$, ensuring interpretability and tree-based search.

# 4 $\alpha$-CFG-SEM-$k$ GUIDED SEARCH FOR HIGH-QUALITY ALPHA FACTORS

## 4.1 TREE-STRUCTURED LINGUISTIC MARKOV DECISION PROCESS

With Definition 4, alpha discovery reduces to: (1) finding a high-quality path from the root to a leaf (i.e., generating a complete alpha), or (2) expanding an intermediate node (e.g., a partially-masked existing factor) into a stronger expression. In the tree-structured search space, each leaf is labeled with the information coefficient (IC) of the resulting alpha, computed from historical market data (Figure 3, Algorithm 1). This *reward* can be backpropagated to ancestors, making each node a *state* with value and each edge an *action*. Thus, the $\alpha$-CFG–guided generation process naturally defines a Markov Decision Process, which we term the Tree-Structured Linguistic MDP (TSL-MDP).

**Definition 5** (TSL-MDP). The alpha discovery process governed by $\alpha$-CFG-Sem-$k$ can be captured by a Tree-Structured Linguistic Markov Decision Process, denoted by TSL-MDP $= \langle S, A, P, R, \gamma \rangle$, where $S$ is the set of partial or complete alpha expressions (states) $s$, each represented by an Abstract Syntax Representation (Definition 6); $A$ is the set of production rules from $\alpha$-CFG-Sem-$k$ defined in Definitions 2 and 3; $P(s' \mid s, a)$ applies production rule $a \in A$ to partial alpha expression $s$, replacing the leftmost nonterminal symbol and yielding expanded alpha expression $s'$; $R(s, a)$ assigns reward only when $a$ produces a complete alpha expression.

**Definition 6** (Abstract Syntax Representation (ASR)). An ASR of an alpha expression is a rooted, ordered tree (shown in zoomed round-boxes in Figure 3) where each node is labeled with an operator (Table 5) and has as many children as required by its arity. Edges represent the application of the parent operator to its child, while leaves are labeled with either a feature (Table 3), a constant (Table 4), or, in partial derivations, a nonterminal symbol.

## 4.2 REINFORCEMENT LEARNING FRAMEWORK

The TSL-MDP is vast, but it is also tree-structured. While classical MCTS can exploit the tree structure, its efficiency breaks down at this linguistic scale. To overcome this, we embed MCTS into a reinforcement learning framework: two neural networks approximate the $\alpha$-CFG production policy and the value of expressions, while a Tree-LSTM encodes the structure of alpha factor. This combination allows MCTS to be guided by learned representations, enabling recursive knowledge acquisition and efficient policy learning over the TSL-MDP (illustrated in Figure 9).

In our RL formulation, the environment is the TSL-MDP, where real-world rewards from market data appear only at leaf nodes. Starting from the initial state (the start symbol of $\alpha$-CFG), we iteratively construct $j = J$ MCTS and corresponding policies. For instance, at $j = 0$, we first perform $i = I$ rounds of MCTS construction, each consisting of selection, expansion, evaluation, and backpropagation guided by three neural networks. This constructed MCTS has a policy. Then we sample an action from this policy and use it as the action at the root, and move to a new node. Then, at $j = 1$, this new node becomes the root while its siblings and their subtrees are discarded. From this new tree, we run the second $i = I$ rounds of network-guided MCTS construction as above, update the policy, and sample the next action. The process repeats until a complete alpha expression

---

[2]Following formal language theory (Hopcroft & Ullman, 1979), each expression is a small tree; to distinguish it from the overall search tree, we call it an ASR.

is generated. Its reward defines a trajectory, and by collecting many such trajectories, we train the policy and value networks via reinforcment learning (see Algorithm 4 for the pseudocode).

## 4.3 GRAMMAR-AWARE MONTE CARLO TREE SEARCH

Inside the RL framework, for any new root $j$, assume at time step $i$, the MCTS agent $M_i$ has covered a subtree of TSL-MDP. Then it executes the following components (see Figure 4 and Appendix B.3 for details).

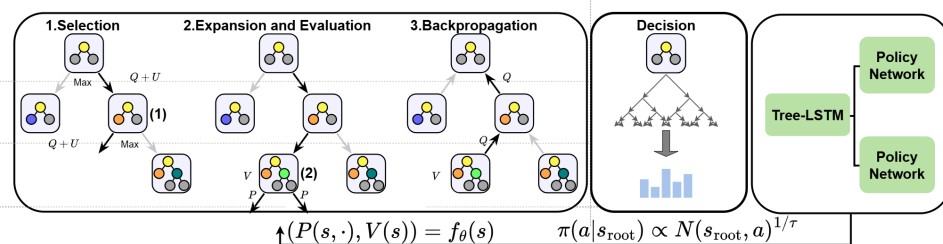

Figure 4: The procedure of grammar-aware MCTS, where value and policy networks are used.

**Selection.** From the root $j$, the MCTS agent $M_i$ repeatedly applies an $\alpha$-CFG production rule to the leftmost nonterminal until reaching a frontier node which has not yet been included in $M_i$. Because TSL-MDP has irregular branching, i.e., varying production rules and shrinking options near the bottom, we adopt a PUCT-style rule (Silver et al., 2017):

$$a^* = \arg\max_a \left( Q(s,a) + c_{\text{puct}} \sqrt{\tfrac{b}{b_{\text{ref}}}} \, P(s,a) \frac{\sqrt{\sum_b N(s,b)}}{1+N(s,a)} \right). \tag{4}$$

**Expansion and Evaluation**. We introduce a Tree-LSTM–based value network $V(s)$ and policy network $P(s,a)$. The selected node is evaluated using the value network $V(s)$. At the frontier node, all the valid production rules are applied, and its valid child nodes are attached. The production rule distribution follows policy $P(s)$, which is the output of policy network.

**Backpropagation**. The evaluation result $V(s)$ is backpropagated along the selection path, updating $Q(s,a)$ and visit counts $N(s,a)$. Iterating these steps allow agent $M_i$ cover more and more nodes in the TSL-MDP (Algorithm 3).

## 4.4 SYNTAX REPRESENTATION LEARNING

**Neural Network Design.** The main challenge in TSL-MDP is its vast state space: we must evaluate both partial/complete alpha expressions and policies for expanding them. Since each state has an ASR (Definition 6), we employ syntax-aware representation learning that directly encodes structure and semantics, avoiding costly full simulations in classic MCTS. Moreover, due to the symmetry of some operators (operands are exchangeable), there are large scale of isomorphic factor expressions (defined in Definition 7) in TSL-MDP. Syntax-aware representation learning is suitable for addressing these redundancies because it directly encodes the ASR rather than linear sequence. Specifically, we use a Tree-LSTM (Tai et al., 2015) with a policy head and a value head (details in Appendix E).

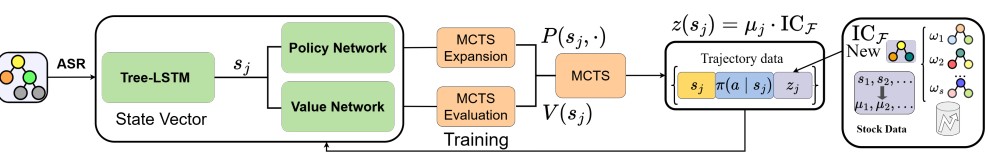

Figure 5: ASR-based representation learning scheme for grammar-aware MCTS.

As shown in Figure 5, the Tree-LSTM recursively aggregates information, producing a fixed-dimensional state vector for each ASR. Then this vector is fed into two networks: (1) the policy network, which outputs probabilities over valid production rules to guide expansion; and (2) the value network, which outputs a scalar for direct use in MCTS evaluation.

**Train the Networks.** We jointly train the policy and value networks using Tree-LSTM representations of TSL-MDP states. In the first round, both networks are randomly initialized: the value network guides MCTS evaluation, and the policy network guides MCTS expansion. Based on these, a full MCTS is constructed, providing an initial alpha generation policy. This policy is then used (i) back to train the policy network and (ii) to sample complete alpha expressions, whose IC values (from market data) supervise the value network. In subsequent rounds, the updated networks guide new MCTS constructions, and the process repeats until enough alphas have been sampled.

Since the final objective is the composite factor $IC_{\mathcal{F}}$ (Appendix B.1), generating expressions structurally similar to existing ones reduces pool diversity and weakens performance. To mitigate this, we introduce a normalized structural similarity measure $\text{sim}(\cdot, \cdot)$, computed via maximum common subtree matching (Sager et al., 2006) between the ASR $f_j$ of $s_j$ and any existing $f_t \in \mathcal{F}$. This similarity penalizes states whose grammar features overlap with $\mathcal{F}$, giving the value target.

$$z(s_j) = \big(1 - \max(0, \max_{f_t \in \mathcal{F}} \text{sim}(f_t, f_j))\big) \cdot IC_{\mathcal{F}}. \tag{5}$$

Training uses triplets $\big(s_j, \pi(a|s_j), z(s_j)\big)$ from each CFG step, where $s_j$ is the Tree-LSTM representation of $j$-th root, $\pi(a|s_j)$ is the MCTS policy distribution, and $z(s_j)$ the value target. Parameters $\theta$ are optimized via the value loss $(z(s_j) - V(s_j))^2$, the policy loss $-\sum_a \pi(a|s_j) \log P(a|s_j)$, and an $\ell_2$ regularization term $c|\theta|^2$. After each round, the updated networks are redeployed, forming an iterative *search–train–search* cycle that progressively improves both efficiency and factor quality.

## 5 EXPERIMENTS

The detailed experiment setting is shown in Appendix (H.1 Data, H.2 Comparison Methods, H.3 Evaluation Metrics). The experimental parameters are provided in G for reproduction. The factor example and the interpretability analysis of generated factors are shown in H.5

**Comparison of Various Generation Approaches** We compared three CFG levels with RPN on CSI 300 and S&P 500 training data to assess how language constraints (Figure 2) affect factor generation. With a pool size of 10 and max length 5, Figure 6 shows training IC across epochs. CFG-S, CFG-SS, and CFG-SSL correspond to $\mathcal{L}_{\text{syn}}$, $\mathcal{L}_{\text{sem}}$, and $\mathcal{L}_{\text{sem}}^{\leq k}$, respectively. Results confirm our analysis (Section 3.3): smaller grammar-defined spaces yield faster convergence and higher-quality factors. Notably, RPN converges to a level close to CFG-SS but more slowly, indicating partial semantic validity yet weaker effectiveness than CFG-SS, highlighting the superiority of our approach.

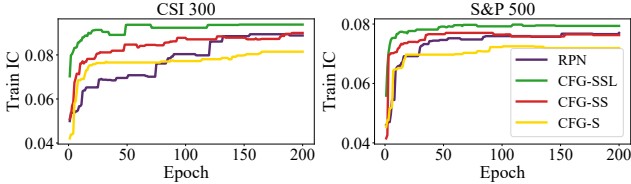

Figure 6: Comparison of training curves of generation methods.

**Comparison of Different Network Architectures** We conducted comparative experiments under different network architectures (Transformer, LSTM, CNN) while keeping other conditions constant. With a pool size of 10 and max length 5, Figure 7 shows training IC across epochs. Results demonstrate the effectiveness and superiority of syntax representation learning. Tree-LSTM not only extracts the structural and semantic information of expressions but also reduces redundancy caused by isomorphic forms (Definition 7).

**Comparison of Multiple Alpha Factor Generation Methods** Under the optimized parameters from the validation dataset experiments (see details in Appendix H.4), we compared our MCTS-

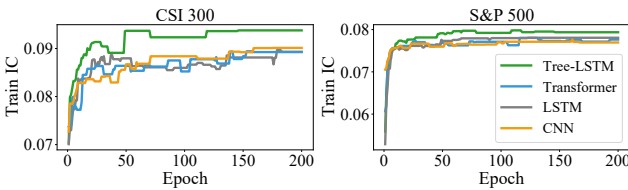

Figure 7: Comparison of training curves of different network architectures.

Table 1: Evaluation metrics comparison of different methods (5 random seeds).

| CSI300 | | | | | | |
|---|---|---|---|---|---|---|
| Method | Rank IC | IC | Rank ICIR | ICIR | Sharpe | Max Drawdown |
| XGBoost | 0.0288 (0.0000) | 0.0326 (0.0000) | 0.2895 (0.0000) | 0.2818 (0.0000) | 0.2853 (0.0000) | **-0.2777 (0.0000)** |
| LightGBM | 0.0539 (0.0029) | 0.0296 (0.0014) | 0.3963 (0.0247) | 0.2649 (0.0395) | 0.2680 (0.0666) | -0.3271 (0.0177) |
| LSTM | 0.0128 (0.0260) | 0.0127 (0.0136) | 0.0896 (0.2064) | 0.1041 (0.1060) | 0.1268 (0.0425) | -0.3542 (0.0240) |
| TCN | 0.0303 (0.0236) | 0.0085 (0.0133) | 0.2726 (0.1855) | 0.0871 (0.1557) | 0.0908 (0.0754) | -0.2988 (0.0191) |
| ALSTM | 0.0138 (0.0076) | 0.0105 (0.0067) | 0.1194 (0.0540) | 0.0950 (0.0550) | 0.1372 (0.1113) | -0.3475 (0.0501) |
| Transformer | 0.0423 (0.0133) | 0.0248 (0.0132) | 0.3759 (0.0697) | 0.2457 (0.0971) | 0.1699 (0.1105) | -0.3365 (0.0377) |
| gplearn | 0.0706 (0.0119) | 0.0440 (0.0139) | 0.4695 (0.1164) | 0.3478 (0.1397) | 0.2062 (0.2346) | -0.3854 (0.0324) |
| AlphaQCM | 0.0811 (0.0046) | 0.0525 (0.0048) | 0.5334 (0.0296) | 0.3874 (0.0121) | 0.4363 (0.0610) | -0.3605 (0.0339) |
| RPN+PPO(AlphaGen) | 0.0837 (0.0070) | 0.0477 (0.0086) | 0.5724 (0.0343) | 0.3531 (0.0574) | 0.4978 (0.1478) | -0.3497 (0.0423) |
| **Ablation Studies** | | | | | | |
| RPN+MCTS | 0.0710 (0.0031) | 0.0500 (0.0026) | 0.5577 (0.0292) | 0.4285 (0.0293) | 0.5639 (0.1050) | -0.3201 (0.0613) |
| CFG-S+MCTS | 0.0745 (0.0052) | 0.0487 (0.0036) | 0.5125 (0.0467) | 0.3974 (0.0367) | 0.4852 (0.1320) | -0.3475 (0.0414) |
| CFG-SS+MCTS | 0.0770 (0.0044) | 0.0512 (0.0015) | 0.5593 (0.0340) | 0.4369 (0.0301) | 0.5801 (0.1169) | -0.3039 (0.0206) |
| CFG-SSL+MCTS(AlphaCFG) | **0.0865 (0.0060)** | **0.0577 (0.0029)** | **0.6036 (0.0537)** | **0.4505 (0.0249)** | **0.6459 (0.0612)** | -0.2963 (0.0289) |

| S&P500 | | | | | | |
|---|---|---|---|---|---|---|
| Method | Rank IC | IC | Rank ICIR | ICIR | Sharpe | Max Drawdown |
| XGBoost | 0.0140 (0.0000) | 0.0104 (0.0000) | 0.1535 (0.0000) | 0.1456 (0.0000) | 0.5883 (0.0000) | -0.2543 (0.0000) |
| LightGBM | 0.0078 (0.0021) | 0.0220 (0.0032) | 0.0860 (0.0269) | 0.2072 (0.0229) | 0.5852 (0.0547) | -0.2047 (0.0128) |
| LSTM | 0.0131 (0.0077) | 0.0219 (0.0040) | 0.1157 (0.0786) | 0.1847 (0.0419) | 0.5601 (0.0546) | -0.2345 (0.0142) |
| TCN | 0.0198 (0.0040) | 0.0166 (0.0020) | 0.1358 (0.0190) | 0.1340 (0.0133) | 0.4973 (0.0271) | -0.2396 (0.0175) |
| ALSTM | 0.0202 (0.0028) | 0.0268 (0.0039) | 0.1569 (0.0344) | 0.1993 (0.0391) | 0.4441 (0.0397) | -0.2418 (0.0109) |
| Transformer | 0.0106 (0.0049) | 0.0185 (0.0036) | 0.0828 (0.0433) | 0.1806 (0.0361) | 0.5979 (0.1163) | -0.2512 (0.0070) |
| gplearn | 0.0130 (0.0122) | 0.0322 (0.0110) | 0.0812 (0.0643) | 0.1877 (0.0437) | 0.8241 (0.1814) | -0.2456 (0.0434) |
| AlphaQCM | 0.0178 (0.0055) | 0.0384 (0.0056) | 0.1149 (0.0381) | 0.2527 (0.0336) | **1.0566 (0.0756)** | -0.2105 (0.0273) |
| RPN+PPO(AlphaGen) | 0.0149 (0.0055) | 0.0342 (0.0050) | 0.1045 (0.0364) | 0.2420 (0.0296) | 0.8271 (0.1421) | -0.2559 (0.0242) |
| **Ablation Studies** | | | | | | |
| RPN+MCTS | 0.0309 (0.0054) | 0.0385 (0.0031) | 0.2447 (0.0234) | 0.3308 (0.0344) | 0.7992 (0.0854) | -0.1957 (0.0140) |
| CFG-S+MCTS | 0.0111 (0.0017) | 0.0272 (0.0047) | 0.0913 (0.0087)) | 0.2335 (0.0356) | 0.8046 (0.0322) | -0.2286 (0.0186) |
| CFG-SS+MCTS | 0.0265 (0.0011) | 0.0413 (0.0030) | 0.2075 (0.0108) | 0.3360 (0.0162) | 0.8315 (0.0855) | -0.2243 (0.0225) |
| CFG-SSL+MCTS(AlphaCFG) | **0.0354(0.0026)** | **0.04573 (0.0034)** | **0.2958(0.0154)** | **0.4099 (0.0230)** | 0.8473 (0.0483) | **-0.1942 (0.0126)** |

based methods (CFG-S, CFG-SS, CFG-SSL and RPN) against existing factor mining methods or prediction models (formulaic: Alphagen, AlphaQCM, GPlearn; ML-based: XGBoost, LightGBM, LSTM, ALSTM, TCN, Transformer). The experiments were conducted separately on the CSI 300 index and the S&P 500 constituents testing data for correlation metrics and backtesting metrics. Notably, the backtesting metrics are obtained based on a single top-$k$/drop-$n$ strategy to conduct simulated trading based on real stock data (detailed at Appendix H.3). The evaluation metrics results are shown in Table 1. In order to demonstrate the trading performance, we calculated the cumulative returns for different methods and obtained Figure 8.

Our method performed the best in all correlation metrics which are directly related to the optimization target IC. Ablation experiments also demonstrated the irreplaceable role of three constraints: syntax, semantic and limited-length. Furthermore, the formulaic factor mining methods generally outperformed the machine learning methods that directly predict stocks in correlation metrics, which proves the potential value of this type of method in quantitative trading.

Although our method does not directly optimize for any one of the backtest metrics, our method still achieves a significant advantage in the MaxDD and Sharpe. What's more, compared with other methods, our method achieves the highest profit.

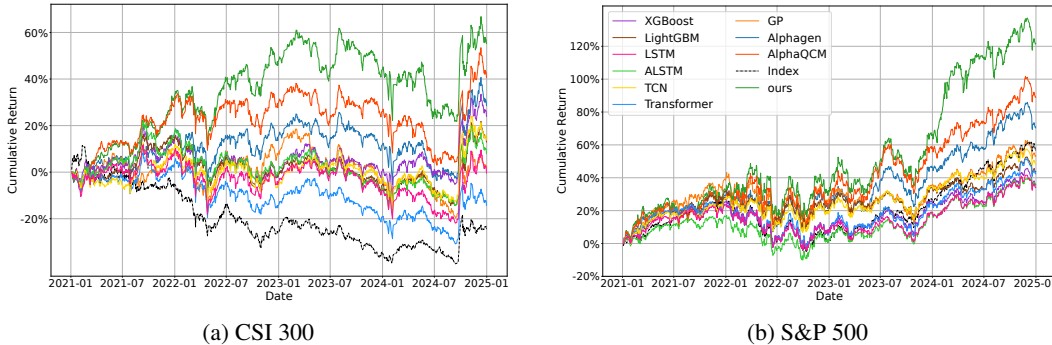

(a) CSI 300                 (b) S&P 500

Figure 8: Cumulative return comparison in simulated trading (the Index in the two figures represents the CSI 300 Index and the S&P 500 Index, respectively)

**Improving traditional alpha factors** In addition to directly mining composite factors, our CFG-SSL+MCTS framework can also refine and strengthen existing classic interpretable factors: We selected a set of factors that have become ineffective but retain financial theoretical interpretability from the Guotai Junan 191 Factor Library (Team, 2017) and the Alpha101 Factor Library (Kakushadze, 2016). Specifically, factors from the Guotai Junan 191 library were improved using the CSI 300 dataset, while those from the Alpha101 library were improved using the S&P 500 dataset. By masking some operators and operands while preserving the left-side structure not exceeding half of the original factor length, we improved these classic factors with the single-factor reward as the optimization objective (blue path in Figure 3). As shown in Table 2, our framework effectively enhances the predictive strength of many classic factors—the absolute IC values are consistently improved on the test sets.

Table 2: Refinement Results: Test Set IC Before and After Applying AlphaCFG.

| GTJA191 | | Alpha101 | |
|---|---|---|---|
| *Original:* open/Ref(close,1)-1 | 0.00185 | *Original:* -Corr(open,volume,10) | 0.00271 |
| *Improved:* open/0.1-Cov(volume,high,20) | 0.04279 | *Improved:* Corr(open,Log(\|open\|),40)·CSRank(high) | 0.02934 |
| *Original:* Mean(close,6)-close | 0.00482 | *Original:* -Rank(CSRank(low),9) | 0.01031 |
| *Improved:* Mean(Cov(vwap,volume,20)/(-0.01),20)/0.05 | 0.04262 | *Improved:* Rank(CSRank(CSRank(Sign(vwap))),30)·CSRank(high) | 0.02944 |
| *Original:* close-Ref(close,5) | 0.00495 | *Original:* Pow(high·low,0.5)-vwap) | 0.00112 |
| *Improved:* close-Greater(-0.1,Cov(volume,\|vwap\|,30)) | 0.03872 | *Improved:* Pow(CSRank(\|open\|)·open,CSRank(close))-vwap | 0.03126 |

## 6 CONCLUSION

CFG is a foundational grammar in computer science and linguistics. Our automated AlphaCFG captures alpha factors' syntactic validity and financial interpretability, provides a recursive syntax-tree structure for alphas, and enables designing a framework integrating reinforcement learning and neural MCTS. Future work might incorporate richer semantic constraints to further enhance the interpretability of generated factors, and use diversified optimization objectives such as turnover and risk-adjusted return beyond IC alone.

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

# A TABLES

Table 3: Stock Feature Variables

| Feature | Description |
| --- | --- |
| open | Opening price |
| high | Highest price |
| low | Lowest price |
| close | Closing price |
| volume | Trading volume |
| vwap | Volume Weighted Average Price (VWAP) |

Table 4: Constant Parameters

| Nonterminal | Values |
| --- | --- |
| Constant | $-0.1, -0.05, -0.01, 0.01, 0.05, 0.1$ |
| Num | $20, 30, 40$ |

Table 5: Formulaic Alpha Factor Operators in Our Framework (the BinaryOp in Formula (3) does not distinguish whether it is symmetric)

| Operator | Type | Description |
| --- | --- | --- |
| $\text{Abs}(x)$ | **Unary** | Absolute value, $|x|$. |
| $\text{Sign}(x)$ | **Unary** | Returns the sign of $x$: 1 for positive, -1 for negative, 0 for zero. |
| $\text{Log}(x)$ | **Unary** | Natural logarithm, $\log(x)$. |
| $\text{Add}(x, y)$ | **Binary** | Addition, $x + y$. |
| $\text{Mul}(x, y)$ | **Binary** | Multiplication, $x \cdot y$. |
| $\text{Greater}(x, y)$ | **Binary** | Returns the larger of two values: $\max(x, y)$. |
| $\text{Less}(x, y)$ | **Binary** | Returns the smaller of two values: $\min(x, y)$. |
| $\text{Div}(x, y)$ | **Binary-Asym** | Division, $x/y$. |
| $\text{Pow}(x, y)$ | **Binary-Asym** | Exponentiation, $x^y$. |
| $\text{Sub}(x, y)$ | **Binary-Asym** | Subtraction, $x - y$. |
| $\text{CSRank}(x)$ | **Rolling** | Cross-sectional ranking (normalizes the rank of $x$ across all stocks on the same day). |
| $\text{Rank}(x, t)$ | **Rolling** | Time-series ranking of $x$ over the past $t$ days. |
| $\text{WMA}(x, t)$ | **Rolling** | Weighted moving average with weights decaying over time. |
| $\text{EMA}(x, t)$ | **Rolling** | Exponential moving average with recursive smoothing. |
| $\text{Ref}(x, t)$ | **Rolling** | Value of $x$ from $t$ days ago. |
| $\text{Mean}(x, t)$ | **Rolling** | Mean of $x$ over the past $t$ days, $\frac{1}{t} \sum_{i=0}^{t-1} x_{-i}$. |
| $\text{Sum}(x, t)$ | **Rolling** | Sum of $x$ over the past $t$ days, $\sum_{i=0}^{t-1} x_{-i}$. |
| $\text{Std}(x, t)$ | **Rolling** | Standard deviation of $x$ over the past $t$ days. |
| $\text{Var}(x, t)$ | **Rolling** | Variance of $x$ over the past $t$ days. |
| $\text{Skew}(x, t)$ | **Rolling** | Skewness (measure of asymmetry) of $x$ over the past $t$ days. |
| $\text{Kurt}(x, t)$ | **Rolling** | Kurtosis (measure of tail thickness) of $x$ over the past $t$ days. |
| $\text{Max}(x, t)$ | **Rolling** | Maximum value of $x$ over the past $t$ days. |
| $\text{Min}(x, t)$ | **Rolling** | Minimum value of $x$ over the past $t$ days. |
| $\text{Med}(x, t)$ | **Rolling** | Median of $x$ over the past $t$ days. |
| $\text{Mad}(x, t)$ | **Rolling** | Mean absolute deviation, $\frac{1}{t} \sum_{i=0}^{t-1} |x_{-i} - \bar{x}|$. |
| $\text{Delta}(x, t)$ | **Rolling** | Difference, $x - \text{Ref}(x, t)$. |
| $\text{Cov}(x, y, t)$ | **PairedRolling** | Covariance between $x$ and $y$ over the past $t$ days. |
| $\text{Corr}(x, y, t)$ | **PairedRolling** | Pearson correlation coefficient between $x$ and $y$ over the past $t$ days. |

Table 6: Length increments $\Delta k$ for each production rule.

| Production Rules | $\Delta k$ |
|---|---|
| Expr $\rightarrow$ Feature | 0 |
| Num $\rightarrow$ 20 . . . | 0 |
| Constant $\rightarrow$ $-0.01$ . . . | 0 |
| Expr $\rightarrow$ UnaryOp(Expr) | 1 |
| Expr $\rightarrow$ BinaryOp(Expr, Expr) | 2 |
| Expr $\rightarrow$ BinaryOp(Expr, Constant) | 2 |
| Expr $\rightarrow$ BinaryOp_Asym(Constant, Expr) | 2 |
| Expr $\rightarrow$ RollingOp(Expr, Num) | 2 |
| Expr $\rightarrow$ PairedRollingOp(Expr, Expr, Num) | 3 |

# B  ALGORITHMS

## B.1  LINEAR COMBINATION ALPHA FACTOR ALGORITHM

The linear combination factor model is defined as

$$c(X; F, w) = \sum_{j=1}^{n} w_j f_j(X) = y, \qquad (6)$$

where $F = \{f_1, \ldots, f_n\}$ denotes the set of factors, $w = \{w_1, \ldots, w_n\}$ are the weights of factors in linear combination , $X$ represents the input stock feature data, and $y$ is the combined output. The optimization is conducted by minimizing the loss function

$$L(w) = \frac{1}{T} \sum_{t=1}^{T} \|y_t - r_t\|^2 \qquad (7)$$

where $r_t$ is the actual stock return, and $y_t$ is the alpha value of linear combination factor.

---

**Algorithm 1** Incremental Combination Model Optimization

---

**Require:** Alpha set $F = \{f_1, \cdots, f_n\}$, weights $w = \{w_1, \cdots, w_n\}$, new alpha $f_{\text{new}}$
**Ensure:** Optimal alpha subset $F^* = \{f'_1, \cdots, f'_n\}$, optimal weights $w^* = \{w'_1, \cdots, w'_n\}$, $\text{IC}_{\mathcal{F}}$
1: $F \leftarrow F \cup \{f_{\text{new}}\}; \quad w \leftarrow w \,\|\, \text{rand}()$
2: **for** $i \leftarrow 1$ **to** num_gradient_steps **do**
3:     Calculate $L(w)$ according to Eq. (7)
4:     $w \leftarrow \text{GradientDescent}(L(w))$
5: **end for**
6: $p \leftarrow \arg\min_i |w_i|$
7: $F \leftarrow F \setminus \{f_p\}; \quad w \leftarrow w \setminus \{w_p\}$
8: Compute the combination IC: $\text{IC}_{\mathcal{F}} \leftarrow \text{IC}(F, w)$
9: **return** $F, w, \text{IC}_{\mathcal{F}}$

---

## B.2  LENGTH CONTROL OF SEMANTIC INTERPRETABLE ALPHA FACTOR GENERATOR

Following the intuition of grammar-constrained generation (Jin et al., 2018), we introduce a $k$-*bounded constraint* to explicitly limit expression length. The mechanism maintains a counter $k$ for the partial length of the expression and enforces a maximum threshold $K$. Each production rule has a predefined increment $\Delta k$, representing its contribution to the expression length(see Table 6 for details). A rule is applied only if $k + \Delta k \leq K$, thereby guaranteeing that each expansion step remains within the feasible bound. By integrating this length-aware constraint into the derivation procedure, we obtain a bounded variant of $\alpha$-CFG-Sem, denoted as $\alpha$-CFG-Sem-K. The procedure is described in Algorithm 2.

---

**Algorithm 2** $\alpha$-CFG-Sem-$k$

---

**Require:** Grammar $G = (\mathcal{N}, \mathcal{T}, \mathcal{P}, \mathcal{S})$,; max length $K$; rule increments $\Delta k : \Gamma \to \beta$
**Ensure:** Prefix expression tree $T$
 1: $T \leftarrow$ single-node tree with root $S$
 2: $k \leftarrow 0$
 3: **while** $T$ contains a nonterminal node **do**
 4:     $u \leftarrow$ first nonterminal node in pre-order traverse
 5:     $\mathcal{A} \leftarrow \{l \in \mathcal{P}$ applicable to $u$ and $k + \Delta k(l) \leq K\}$
 6:     choose $l : \Gamma \to \beta$ from $\mathcal{A}$
 7:     replace node $u$ with children realizing $\alpha$
 8:     $k \leftarrow k + \Delta k(l)$
 9: **end while**
10: **Return** $T$

---

## B.3 ALGORITHM OF FOUR STAGES OF MCTS

---

**Algorithm 3** Grammar-aware MCTS with Branch-adapted PUCT

---

**Require:** Root state $s_{\text{root}}$, policy-value network $\theta$, iteration count $I$
**Ensure:** Improved policy $\pi(a|s_{\text{root}})$
 1: **for** $i = 1$ to $I$ **do**
 2:     $s \leftarrow s_{\text{root}}$
 3:     Initialize empty list of traversed edges $E \leftarrow [\,]$
 4:     **while** $s$ is not fully expanded **do**
 5:         $b \leftarrow$ number of valid actions from $s$
 6:         $a^* \leftarrow \arg\max_a \left[ Q(s, a) + c_{\text{puct}} \cdot \sqrt{\frac{b}{b_{\text{ref}}}} \cdot P(s, a) \cdot \frac{\sqrt{\sum_b N(s,b)}}{1 + N(s,a)} \right]$         $\triangleright$ Selection
 7:         Append $(s, a^*)$ to $E$
 8:         $s \leftarrow \text{apply}(s, a^*)$
 9:     **end while**
10:     $s_L \leftarrow s$
11:     $(P(s_L, \cdot), V(s_L)) \leftarrow f_\theta(s_L)$         $\triangleright$ Expansion and Evaluation
12:     Expand $s_L$ with $P(s_L, \cdot)$
13:     **for all** $(s, a) \in E$ **do**
14:         $N(s, a) \leftarrow N(s, a) + 1$         $\triangleright$ Backpropagation
15:         $Q(s, a) = \frac{1}{N(s,a)} \sum_{s'|s,a \to s'} V(s')$
16:     **end for**
17: **end for**
18: $\pi(a \mid s_{\text{root}}) = \frac{N(s_{\text{root}}, a)^{1/T}}{\sum_{b \in A(s_{\text{root}})} N(s_{\text{root}}, b)^{1/T}}$
19: **Return** $\pi(a|s_{\text{root}})$

---

Assume that at a certain iteration $i$, our MCTS has already explored a portion of the TSL-MDP, denoted by an agent $M_i$. This agent corresponds to a subtree of the large TSL-MDP, sharing the same root, and $M_i$ has obtained policy for this partial subtree. For example, at simulation $M_i$, the subtree agent $M_i$ shown on the left in Figure 4 has already been explored. This subtree starts as only a root when $i = 0$, and is intended to expand toward the full TSL-MDP tree as $i$ increases, eventually reaching iteration $i = I$.

**Selection.** First, within $M_i$, starting from root of the subtree, the MCTS agent repeatedly selects an $\alpha$-CFG production rule at each incomplete alpha expression (each round-box node), and replaces its leftmost nonterminal symbol (the dark black arrows in Figure 4), which goes to a new incomplete alpha expression (a child round-box node). This repeats until it reaches a "frontier" alpha expression that has a child not yet included in $M_i$ (e.g., node (1) in Figure 4).

The TSL-MDP has two key features: (1) different nonterminal symbols have different numbers of production rules, and (2) the number of valid production rules decreases sharply near the bottom of

the search tree due to the length control in B.2. To address this, we adopt a production rule selection function analogous to PUCT (Silver et al., 2017).

$$a^* = \arg\max_a \left( Q(s,a) + c_{\text{puct}} \cdot \sqrt{\tfrac{b}{b_{\text{ref}}}} \cdot P(s,a) \cdot \frac{\sqrt{\sum_b N(s,b)}}{1 + N(s,a)} \right), \tag{8}$$

Here, $Q(s,a)$ is the value of selecting production rule $a$ for formula $s$, and $P(s,a)$ is the probability of selecting $a$ under $s$. $b$ is the number of branches at the current depth, and $b_{\text{ref}}$ is the branch balance constant (defined by the maximum number of branches) Eq. (8) balances irregular branching through the adaptive term $\sqrt{b/b_{\text{ref}}}$: smaller branching factors emphasize exploitation, while larger ones promote broader exploration.

**Expansion.** After finding such a frontier alpha expression node, the MCTS agent will execute a certain production rule on it, generating a new alpha expression which has not yet been covered by $M_i$ (e.g., round-box node (2) in Figure 4), and also attaching all the corresponding possible production rules to this new alpha expression (e.g., the two arrows attached to node (2)). The probabilities for executing available production rules for expression $s$ follow the distribution $P(s)$.

**Evaluation.** Since the newly expanded alpha expression is at the head of the current agent $M_t$ and remains incomplete, the existing policy cannot assess its quality. Thus, MCTS requires a method to evaluate it. Given the vastness of the TSL-MDP, traditional simulation-based evaluation is infeasible. Moreover, as shown in Definition 6, the expressions at any state in TSL-MDP are small tree structures (i.e., the small trees inside each round-box in Figure 4). Therefore, in the next section, we design a Tree-LSTM–based representation learning method to construct a value network for $V(s)$, as well as a policy network $P(s,a)$ over any expression.

**Backpropagation.** The result $V(s)$ of evaluation is backpropagate from the path of selection (the path directed by black arrow in the third tree of Figure 4). Mean value of each eadge in the path is updated by $V(s)$ and visit count $N(s,a)$ of each eadge in the path increases by one.

The MCTS agent $M_i$ executes the above procedures at each iteration $i$ (Algorithm 3 shows the procedure of MCTS search.). Since one node is expanded at each step, the MCTS agent $M_i$ will eventually cover enough nodes and edges of the TSL-MDP. The resulting search assigns a *basic value* to every node and obtain a basic policy for the TSL-MDP, which two can be used to further optimize the policy.

## C  Reinforcement Learning Framework

We present pseudo-code of MCTS combined with reinforcement learning method (Algorithm 4). This is a reinforcement learning-based factor mining method designed to automatically discover a combination of factors from stock market data that can effectively predict stock returns. Specifically, the algorithm initializes a set of factors, their corresponding weights, and a policy-value network. In the process of obtaining data through reinforcement learning, it employs a MCTS policy to generate actions for each state, thereby constructing a multi-step factor generation path. The final state of the path is parsed into a computable alpha expression, evaluated using the $IC_\mathcal{F}$ as the reward signal. The reward is given along with the optimization of the factor combination $\mathcal{F}$. The actual value for each step along the path, denoted as $z_t$ is computed based on $IC_\mathcal{F}$ and the similarity between the newly generated factor and existing ones, following the formulation in Equation (5) in Section 4.4.

After generating multi-step factor paths in each iteration, the policy and value networks are trained using the collected path data $(s_j, \pi(a|s_j), z_j)$ stored in a replay buffer, where $s_j$ is the state vector encoded by TreeLSTM, $\pi(a|s_j)$ is the policy from MCTS, and $z_t$ is shown above. After training, the networks are redeployed to guide a new round of search. Through iterative training and exploration, the IC of the learned factor combination is progressively improved. The algorithm outputs the final optimized factor combination set along with its corresponding weights when the IC shows no more significant improvement.

---

**Algorithm 4** Alpha Mining via reinforcement learning

---

**Require:** Stock trend dataset $Y = \{y_t\}$
**Ensure:** Optimal alpha subset $F^* = \{f'_1, \ldots, f'_k\}$, optimal weights $w^* = \{w'_1, \ldots, w'_k\}$
1: Initialize $F$ and $w$
2: Initialize policy-value network $\theta$ and replay buffer $D$
3: **for** each epoch **do**
4:     **for** each factor path search **do**
5:         $E \leftarrow [\,]$
6:         **for** $j = 0$ to $J$ **do**
7:             Append $s_t$ to $E$
8:             $s_{root} \leftarrow s_j$
9:             $\pi(a|s_j) \leftarrow \pi(a|s_{\text{root}})$          ▷ $\pi(a \mid s_{\text{root}})$ is obtained based on Algorithm 3
10:             $a_j \sim \pi(a \mid s_j)$
11:             $s_{j+1} \leftarrow [s_j, a_j]$
12:         **end for**
13:         $f_j \leftarrow \text{parse}(s_{K-1})$          ▷ parse the ASR into a computable alpha expression
14:         Reward $IC_\mathcal{F}$ is obtained using Algorithm 1 by inputting $f_{\text{new}}$, $F^*$ and $w^*$
15:         **for** $j = 0$ to $J$ **do**
16:             $z(s_j) = (1 - \max(0, \max_{f_t \in F} \text{sim}(f_t, f_j))) \cdot IC_\mathcal{F}$
17:             $D \leftarrow D \cup \{(s_j, \pi(a|s_j), z_j)\}$
18:         **end for**
19:     **end for**
20:     **for** each gradient step **do**
21:         Use batch $B \subset D$ to do gradient descent
22:         $L_\theta = (z(s_t) - V_\theta(s_t))^2 - \sum_a \pi(a \mid s_t) \log P_\theta(a \mid s_t) + c\|\theta\|^2$
23:         $\theta_{e+1} = \theta_e - \eta \cdot \nabla_\theta L(\theta_e)$
24:     **end for**
25: **end for**
26: **return** $F^*$, $w^*$

---

The overall workflow of this algorithm is illustrated in Figure 9 in the following page, while a specific illustration of its MCTS component Algorithm 3 is in Figure 4, and the illustration of its neural network part is in Figure 5.

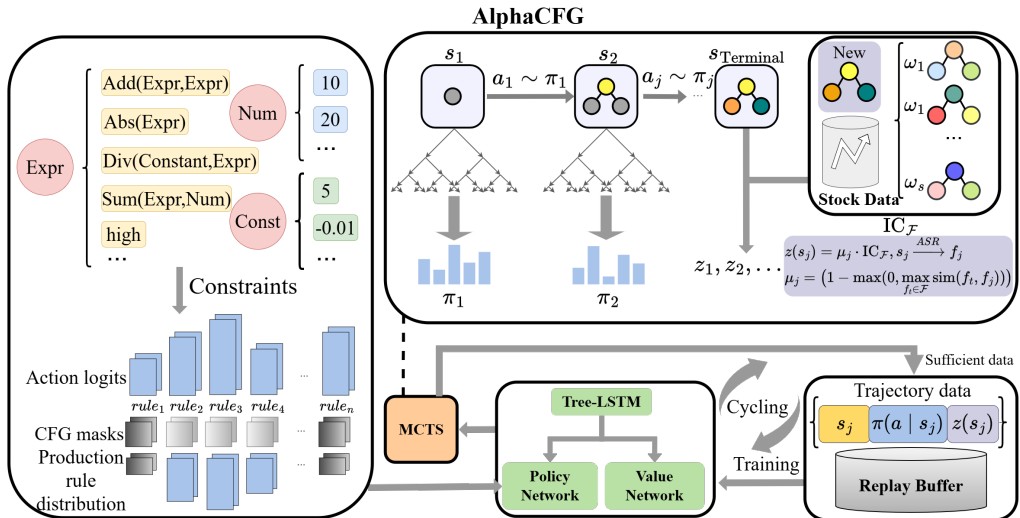

Figure 9: The overall framework of AlphaCFG.

# D  SEARCH SPACE COMPLEXITY

To compare the sizes of expression search spaces under different generation methods, we study three methods from a combinatorial perspective: (i) a purely exponential baseline (arbitrary combination of all symbols corresponding to $\Sigma^*$); (ii) $\alpha$-CFG-Syn (corresponding to $\mathcal{L}_{\text{syn}}$); (iii) $\alpha$-CFG-Sem (corresponding to $\mathcal{L}_{\text{syn}}$). All three methods share the same parameter sets (operator types, number of features, constants, etc.), but progressively impose stricter constraints, resulting in smaller search spaces.

We set the following notation: the size of the unary operator set is $|U|$, the size of the binary operator set is $|B|$, the size of the asymmetric binary operator set is $|B_{\text{asym}}|$, the size of the rolling operator set is $|R|$, the size of the paired rolling operator set is $|R_{\text{pair}}|$, the number of features is $|\mathcal{F}|$, the number of constant parameters is $|\mathcal{C}|$, and the number of rolling-window parameters is $|\mathcal{N}|$.

## D.1  UNSTRUCTURED SPACE $\Sigma^*$

The method of arbitrary symbol combination (referred to ) takes one symbol equally at each step from all available symbols. Let the total number of symbols be:

$$r = |\mathcal{F}| + |\mathcal{C}| + |\mathcal{N}| + |U| + |B| + |B_{\text{asym}}| + |R| + |R_{\text{pair}}|.$$

Then the number of sequences of length $n$ is $r_n = r^n$, and the cumulative size is $\sum_{i \leq n} r_i = \Theta(r^n)$.

## D.2  SYNTACTICALLY LEGAL SPACE $\mathcal{L}_{\text{syn}}$

We introduce syntax constraints to ensure that generated expressions are all syntactically valid. We consider the grammar $\alpha$-CFG-Syn:

Expr $\rightarrow$ UnaryOp($E$) | BinaryOp($E, E$) | RollingOp($E, E$) | PairedRollingOp($E, E, E$) | TermSyb.

Let $h_n$ be the number of valid expressions of length $n$. The terminal set size is: $T = |\mathcal{F}| + |\mathcal{C}| + |\mathcal{N}|$.

Define operator cardinalities: $U = |U|, Q = |B| + |B_{\text{asym}}|, R = |R|, P = |R_{\text{pair}}|$, respectively(The meanings of the notations are as shown in D).

The recurrence formula is: $h_1 = T$, and for $n \geq 2$:

$$h_n = U h_{n-1} + (Q + R) \sum_{i=1}^{n-2} h_i\, h_{n-1-i} + P \sum_{\substack{i+j+k=n-1 \\ i,j,k \geq 1}} h_i h_j h_k.$$

The subsequent derivation of an explicit form from this recurrence becomes rather cumbersome. Since the technical steps mirror the usual treatment of general cubic functional equations, we omit the full derivation here.

## D.3  SEMANTICALLY LEGAL SPACE $\mathcal{L}_{\text{sem}}$

$\alpha$-CFG-Sem introduces more constraints on constants, argument types, and rolling windows:

Expr $\rightarrow$ Feature | UnaryOp(Expr)
   | BinaryOp(Expr, Expr) | BinaryOp(Expr, Constant)
   | BinaryOp_Asym(Constant, Expr) | RollingOp(Expr, Num)
   | PairedRollingOp(Expr, Expr, Num),

Num $\rightarrow$ 20 | $\cdots$,  Constant $\rightarrow$ $-0.01$ | $\cdots$

Let $f_n$ denotes the number of valid expressions of length $n$.

The recurrence formula becomes

$$f_n = |U|\, f_{n-1} \qquad\qquad \text{(unary)}$$

$$+ |B| \sum_{i=1}^{n-2} f_i f_{n-1-i} \qquad \text{(binary)}$$

$$+ |B|\,|\mathcal{C}|\, f_{n-2} \qquad\qquad \text{(binary + right constant)}$$

$$+ |B_{\mathrm{asym}}|\,|\mathcal{C}|\, f_{n-2} \qquad\qquad \text{(asymmetric binary + left constant)}$$

$$+ |R|\,|\mathcal{N}|\, f_{n-2} \qquad\qquad \text{(rolling)}$$

$$+ |R_{\mathrm{pair}}|\,|\mathcal{N}| \sum_{i=1}^{n-3} f_i f_{n-2-i} \qquad \text{(paired rolling)}.$$

The recurrence formula is similar, and compared with $\alpha$-CFG-Syn, recurrence of $\alpha$-CFG-Sem includes more convolution terms and more realistic constraints, providing a more accurate operator usage. In the following, we present the overall analysis.

Because the expression length is unbounded, the search spaces of all three generation methods are infinite. Therefore, the comparison does not concern the total size of each space, but rather the size of the finite subspace consisting of expressions whose length is at most $n$.

For each grammar, the production rules yield a recurrence for the number of expressions of exact length $n$ ) ($r_n, h_n, f_n$), and accumulating these values from $1$ to $n$ gives the size of the corresponding truncated subspace. By computing these cumulative counts and plotting their growth as functions of $n$, we can directly compare how quickly the reachable portions of the three search spaces expand.

### D.4 EMPIRICAL VERIFICATION

Based on the recurrence formulas, We compute the cumulative counts of $\{r_n\}$, $\{h_n\}$, and $\{f_n\}$ for $n = 1 \sim N$, and plot their growth curves to visualize differences between the three methods (shown in Figure 10). Since all three methods yield inherently infinite search spaces, we further design $\alpha$-CFG-Sem-K based on Algorithm 2, which can be seen as the red dotted line in Figure 10. The results are consistent with the analysis in Figure 2, which further strengthens the superiority of our approach in theory.

Figure 10 explains the core of the superiority of our method: By introducing constraints of syntax and semantics, We get an infinite set containing only valid factors. In actual factor search tasks, we cannot exhaust this space that exploring a finite subset is realistic. Therefore, We utilize the recursive feature of CFG and further designed $\alpha$-CFG-Sem-K capable of generating factors of only a finite length. Ultimately, we reduced the complexity of the search space from an exponential level to a constant level, making this task solvable.

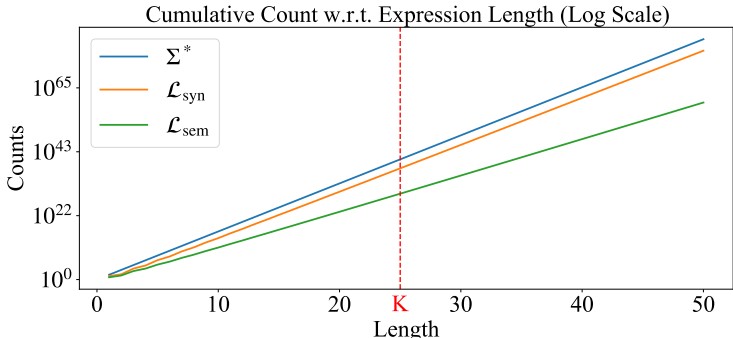

Figure 10: Comparison of cumulative search space sizes of different grammar levels.

# E  DETAILS OF TREE-LSTM

Starting from ASR leaf nodes, the Tree-LSTM recursively aggregates child hidden and cell states through gating (input, forget, output), combining them with the node's input embedding. This bottom-up process continues until the root, yielding a fixed-dimensional state vector that encodes both the syntax and operator-specific dependencies of the entire expression. Thus, the Tree-LSTM transforms variable-sized trees into single vectors while preserving structural and semantic information.

In our $\alpha$-CFG, operators are different: (*i*) symmetric operators, where order is irrelevant, and (*ii*)asymmetrical (order-sensitive) operators, where order must be preserved. Tree-LSTM naturally supports both cases through two variants: the N-ary Tree-LSTM, which uses position-sensitive parameters to encode child order, and the Child-Sum Tree-LSTM, which aggregates child states by their mean to provide order-invariant representations. Based on these, we tailor aggregation strategies: for symmetric binary operators (Expr → BinaryOp(Expr, Expr)) we adopt Child-Sum to avoid redundant encodings; for paired rolling operators (Expr → PairedRollingOp(Expr, Expr, Num)) we first apply unordered aggregation to operands and then use N-ary encoding to incorporate the time-window parameter; and for all other operators we employ standard N-ary encoding. Such operation can address the problem of isomorphic redundancy of alpha factors defined in Definition 7.The resulting tree embeddings are treated as input to be given into the policy and value heads to predict next-rule probabilities and estimated state value.

## E.1  N-ARY TREE-LSTM (POSITION-SENSITIVE)

Let node $j$ have $N$ children with hidden states $\mathbf{h}_1, \ldots, \mathbf{h}_N$, input $\mathbf{x}_j$, output hidden state $\mathbf{h}_j$ and cell state $\mathbf{c}_j$:

$$\mathbf{i}_j = \sigma\left(W^{(i)}\mathbf{x}_j + \sum_{k=1}^{N} U_k^{(i)}\mathbf{h}_k + \mathbf{b}^{(i)}\right)$$

$$\mathbf{f}_{jk} = \sigma\left(W^{(f)}\mathbf{x}_j + U_k^{(f)}\mathbf{h}_k + \mathbf{b}^{(f)}\right), \quad k = 1, \ldots, N$$

$$\mathbf{o}_j = \sigma\left(W^{(o)}\mathbf{x}_j + \sum_{k=1}^{N} U_k^{(o)}\mathbf{h}_k + \mathbf{b}^{(o)}\right)$$

$$\mathbf{u}_j = \tanh\left(W^{(u)}\mathbf{x}_j + \sum_{k=1}^{N} U_k^{(u)}\mathbf{h}_k + \mathbf{b}^{(u)}\right)$$

$$\mathbf{c}_j = \mathbf{i}_j \odot \mathbf{u}_j + \sum_{k=1}^{N} \mathbf{f}_{jk} \odot \mathbf{c}_k$$

$$\mathbf{h}_j = \mathbf{o}_j \odot \tanh(\mathbf{c}_j)$$

## E.2  CHILD-SUM TREE-LSTM

Let node $j$ have a set of children $C(j)$ with hidden states $\mathbf{h}_k$, $k \in C(j)$:

$$\tilde{\mathbf{h}}_j = \frac{1}{|C(j)|} \sum_{k \in C(j)} \mathbf{h}_k$$

$$\mathbf{i}_j = \sigma \left( W^{(i)} \mathbf{x}_j + U^{(i)} \tilde{\mathbf{h}}_j + \mathbf{b}^{(i)} \right)$$

$$\mathbf{f}_{jk} = \sigma \left( W^{(f)} \mathbf{x}_j + U^{(f)} \mathbf{h}_k + \mathbf{b}^{(f)} \right), \quad k \in C(j)$$

$$\mathbf{o}_j = \sigma \left( W^{(o)} \mathbf{x}_j + U^{(o)} \tilde{\mathbf{h}}_j + \mathbf{b}^{(o)} \right)$$

$$\mathbf{u}_j = \tanh \left( W^{(u)} \mathbf{x}_j + U^{(u)} \tilde{\mathbf{h}}_j + \mathbf{b}^{(u)} \right)$$

$$\mathbf{c}_j = \mathbf{i}_j \odot \mathbf{u}_j + \sum_{k \in C(j)} \mathbf{f}_{jk} \odot \mathbf{c}_k$$

$$\mathbf{h}_j = \mathbf{o}_j \odot \tanh(\mathbf{c}_j)$$

# F    CALCULATION OF TREE SIMILARITY

**Definition 7** (Isomorphism of ASR(Tree)). ASR $T_1$ and $T_2$ are isomorphic only if:

1. The label of root nodes must be the same;

2. Recursively check each child node, the labels of the child nodes are equivalent: for asymmetrical operations, the order of the subtrees must be preserved; for symmetrical operations (Binary type operators in Table 5) or partially symmetrical operations (Corr, Cov, where the order of the first two operands' child nodes doesn't matter), the order of the subtrees doesn't matter as long as the operands match;

3. Recursively check that all child nodes and their structures are isomorphic.

Given two alpha factor expresions(partial or completed), they correspond to two ASRs $T_1$ and $T_2$ which are also two trees. Let $\mathrm{Sub}(T)$ denote the set of all subtrees of $T$, where each subtree is induced by a child of node in $T$ along with all its descendant nodes (including the child node itself). Let $N(T)$ denote the total number of subtrees in $T$, recursively defined as:

$$N(T) = 1 + \sum_{c \in \mathrm{Children}(T)} N(c).$$

The normalized similarity between the two ASR is defined as:

$$\mathrm{sim}(T_1, T_2) = \frac{\max_{\substack{t_1 \in \mathrm{Sub}(T_1) \\ t_2 \in \mathrm{Sub}(T_2)}} \mathrm{css}(t_1, t_2)}{\max\left(N(T_1),\ N(T_2)\right)},$$

where the numerator represents the size of the largest isomorphic subtree shared by $T_1$ and $T_2$, i.e., the number of matching nodes in the largest common subtree. Tree isomorphism is defined formally in Definition 7. If no such isomorphic subtree exists, then $\mathrm{css}(t_1, t_2) = 0$.

The denominator $\max(N(T_1), N(T_2))$ corresponds to the number of nodes in the larger of the two trees, serving as an upper bound for the size of any common subtree. Intuitively, it reflects the maximum number of matching nodes that could be achieved if one tree were a subtree of the other, or if the two trees were structurally identical. As such, the denominator defines the *maximum potential scale* of a common subtree, and serves to normalize the matching node count in the numerator. This ensures that the resulting similarity score lies within the standardized range $[0, 1]$, thereby facilitating both quantitative analysis and intuitive comparison of structural similarity between expression trees.

# G  ALPHACFG FRAMEWORK PARAMETER SETTING FOR EXPERIMENT

## G.1  MCTS PARAMETERS

- Exploration Parameter : The exploration-exploitation trade-off parameter in the UCT formula is set to $c = 1$.
- MCTS Simulations : 64 simulations are performed per state.
- MCTS Parallelism: 8 parallel simulations are used to speed up the exploration.
- Eval Batch Size: 2 evaluations using network are carried out simultaneously each time.
- Branch balance coefficient: 40

## G.2  NETWORK ARCHITECTURE

**Feature Extractor (Tree-LSTM)**:

- Embedding Dimension: 128.
- Hidden Size: 128.
- Dropout Rate: 0.1.

**Policy Network**:

- Input: Features extracted by the feature extractor (Tree-LSTM).
- Hidden Layers:
    - Layer 1: Fully connected layer with 128 input features and 64 output features.
    - Layer 2: Fully connected layer with 64 input features and 128 output features (embedding dimension).
- Activation Function: Softmax

**Value Network**:

- Input: Features extracted by the feature extractor (Tree-LSTM).
- Hidden Layers:
    - Layer 1: Fully connected layer with the embedding dimension (128) as input and 64 output features.
    - Layer 2: Fully connected layer with 64 input features and 64 output features.
- Activation Functions: ReLU activation functions applied to the hidden layers.
- Output: A fully connected layer with a single output value without activation function.

## G.3  OPTIMIZER AND TRAINING PARAMETERS

- Optimizer: Adam optimizer with default settings
- Learning Rate: A learning rate of $10^{-4}$.
- Batch Size: 64.
- Number of factor trajectories in an iteration: 100(2*50).
- Training Iterations: 100 iterations.
- Batch Size for Training: 64.
- Replay Buffer Size: 20,000.
- Early Stopping Criteria: Early stopping based on validation performance, with a threshold of 20% iterations without improvement.

# H   More Results of Experiment

We evaluate the proposed framework on both the China A-share and U.S. equity markets. Our experiments are designed to: (1) demonstrate that the proposed context-free grammar provides practical advantages over linear generation methods (e.g., Reverse Polish Notation) for representing and generating alpha factors; (2) validate that the syntax representation learning method using Tree-LSTM to encode state outperforms linear network architectures; (3) evaluate the performance of the grammar-aware discovery framework across multiple metrics in comparison with existing factor-mining methods; (4) assess whether the alpha factors discovered by our model deliver superior trading performance in realistic backtesting scenarios; and (5) examine how our model enhances the performance of existing classical factors.

## H.1   Data

For the A-share market, we adopt the constituent stocks of the CSI 300 index, and for the U.S. market, we use the constituent stocks of the S&P 500 index. The dataset is temporally partitioned into three subsets: the training set (2010-01-01 to 2017-12-31), the validation set (2018-01-01 to 2019-12-31), and the testing set (2021-01-01 to 2024-12-31). To avoid distortions caused by abnormal market volatility and structural irregularities during the COVID-19 pandemic, data from calendar year 2020 are excluded by design. Six raw stock-level features are used as model inputs: {open, close, high, low, volume, vwap}. Formulaic alpha factors are constructed by applying arithmetic operators to these base features under the grammar constraints described earlier. The prediction target for factors is the 20-day forward return, computed using closing prices for both buying and selling, i.e., $R_t^{(20)} = \frac{\text{Ref}(\text{close}, -20)}{\text{close}} - 1$.

## H.2   Comparison Methods

We evaluate three variants of grammar-constrained factor discovery method: (i) **CFG-S** (generation constrained solely by syntactic rules) (ii) **CFG-SS** (generation constrained by both syntactic and semantic rules) (iii) **CFG-SSL** (generation further restricted by a length-bounding mechanism in Algorithm 2). To further validate the grammar effectiveness, we also incorporate Reverse Polish Notation (RPN). (Specifically for **CFG-S**, we constrain the rolling window size to be an integer constant in $\alpha$-CFG-Syn to facilitate smooth training.)

For a broader performance assessment of the entire framework, we compare our method against two state-of-the-art factor mining baselines: AlphaGen (Yu et al., 2023) and AlphaQCM (Zhu & Zhu, 2025). Both employ RPN, with AlphaGen using Proximal Policy Optimization (PPO) and AlphaQCM using distributed reinforcement learning. Additionally, GPlearn (Zhang et al., 2020) is included as a symbolic-regression baseline, which generates formula trees through genetic programming. All of the above factor generation methods optimize the Information Coefficient (IC) of the linear combination of factors.

To further validate our approach, we include several widely used machine learning models as additional baselines: XGBoost (Wang et al., 2023), LightGBM (Bisdoulis, 2024), LSTM (Bhandari et al., 2022), ALSTM (Qin et al., 2017), TCN (Dai et al., 2022), and Transformer (Mozaffari & Zhang, 2024). The hyperparameters of these models are set according to the benchmark configurations provided by Qlib (Yang et al., 2020). To mitigate the impact of randomness, all models are trained and evaluated 5 times with different fixed random seeds.

## H.3   Evaluation Metrics

We evaluate factor effectiveness from two complementary perspectives: correlation metrics, including IC, RankIC, ICIR, and RankICIR, capture the statistical relationship between factors and future returns. Backtesting metrics, which are obtained by investment simulation using a top-k/drop-n strategy (see the next paragraph for details ), including MaxDD and Sharpe, assess the profitability and risk characteristics of factors in simulated trading (see Table 7 for details).

Top-$k$/drop-$n$ strategy is applied to simulate actual trading operations: for each trading day, we first ranked stocks based on their factor prediction scores, then selected the top $k$ stocks from the sorted

list. To balance return potential and trading costs, we adopted an equal-weight allocation approach while limiting daily portfolio adjustments to a maximum of n stocks. In our experiment, we set $k = 60$ and $n = 5$, ensuring sufficient portfolio diversification while controlling transaction costs.

Table 7 provides the specific calculation methods for all evaluation metrics.

| Category | Metric Name | Abbrev. | Formula | Description |
|---|---|---|---|---|
| Correlation Metrics | Information Coefficient | IC | $IC = \rho(\alpha_i, R_i)$ | Pearson correlation between factor values $\alpha_i$ and future returns $R_i$. |
| | Rank Information Coefficient | RankIC | $RankIC = \rho(r(\alpha_i), r(R_i))$ | Spearman correlation after ranking; $r(\cdot)$ is the rank function. |
| | Information Ratio | ICIR | $ICIR = \dfrac{\overline{IC}}{\sigma_{IC}}$ | Ratio of mean IC to its volatility, measuring prediction stability. |
| | Rank Information Ratio | RankICIR | $RankICIR = \dfrac{\overline{RankIC}}{\sigma_{RankIC}}$ | Ratio of mean RankIC to its volatility, evaluating rank correlation stability. |
| Backtesting Metrics | Maximum Drawdown | MaxDD | $MaxDD = \max_t \dfrac{P_{max}(0,t) - P_t}{P_{max}(0,t)}$ | Largest peak-to-trough decline in backtest; $P_t$ is NAV, $P_{max}(0,t) = \max_{s \leq t} P_s$. |
| | Sharpe Ratio | Sharpe | $Sharpe = \dfrac{\mathbb{E}[r_p - r_f]}{\sigma_{r_p}} \times \sqrt{N}$ | Annualized excess return per unit risk; $r_p$: daily return, $r_f$: risk-free rate, $N$: 252 (trading days). |

Table 7: Summary of Evaluation Metrics

### H.4 OPTIMIZATION OF COMBINED FACTOR PARAMETERS ON THE VALIDATION SET

To obtain the optimized combined factor parameters, we conducted experiments on the validation set for two dimensions: *Maximum Length of Individual Factors (Max Length)* and *Factor Pool Size (Pool Size)* (results shown in Figure 11). Specifically, we first fix the maximum length of individual factors and then evaluate the valid IC for different pool sizes {1, 5, 10, 20, 30} to select the optimal pool size. After selecting the optimal pool size under CFG-SSL, we fix it and then explore different values of the maximum length of individual factors {5, 10, 15, 20, 25} to identify the best configuration.

Finally, we obtain the best combined factor parameters:

**CSI 300:**

- RPN+MCTS: Max Length: 10; Pool Size: 20
- CFG+S: Max Length: 10; Pool Size: 20
- CFG+SS: Max Length: 10; Pool Size: 10
- CFG+SSL: Max Length: 10; Pool Size: 10
- RPN+PPO: Max Length: 20; Pool Size: 20

**S&P 500:**

- RPN+MCTS: Max Length: 20; Pool Size: 20
- CFG+S: Max Length: 10; Pool Size: 20
- CFG+SS: Max Length: 10; Pool Size: 20
- CFG+SSL: Max Length: 10; Pool Size: 20
- RPN+PPO: Max Length: 20; Pool Size: 20

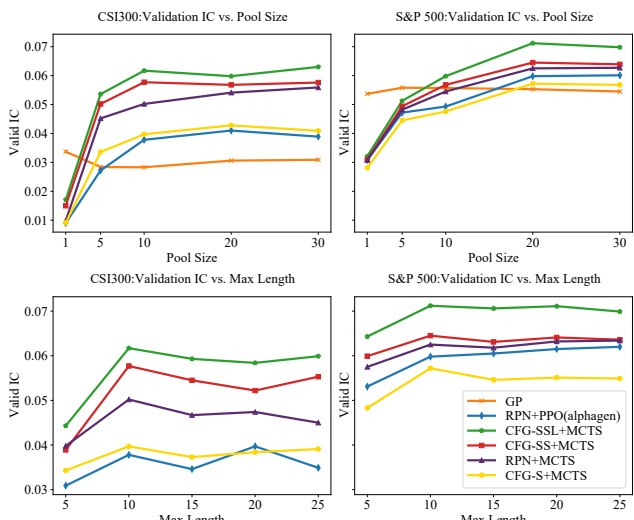

Figure 11: Valid IC of various generation approaches.

The optimization objective of the GP method using a combined model has little effect (the generated combined factors are highly similar), so only the single-factor IC is used as its optimization objective.

### H.5 CASE STUDY OF THE INTERPRETABILITY OF FORMULAIC FACTORS

Table 8 shows an example of alpha factors generated by our framework, tested on the CSI 300 index constituents. The mined factors exhibit strong interpretability grounded in market microstructure theory. For example, the factor Log(|Std((0.05-volume),40)|) measures the volatility of inverse trading volume over a 40-day window. This factor gauges the temporal variability of illiquidity, which may signal market stress or substantial price impact. Another example, Cov(volume,vwap,40), captures the co-movement between trading volume and the volume-weighted average price in past 40 days. A high covariance indicates strong directional consensus, potentially reflecting persistent momentum or, conversely, price reversals.

Table 8: Top 10 Ranked Alphas and Their Weights

| # | Alpha Expression | Weight |
|---|---|---|
| 1 | Mean(Corr(Sum(open,40),(high-volume),20),20) | -0.00889 |
| 2 | volume | -0.01278 |
| 3 | Std(close,40) | 0.01778 |
| 4 | Pow(Med(Cov(high,low,30),30),0.1) | 0.01411 |
| 5 | Delta(Log(|Min(high,30)/0.01|),30) | -0.01649 |
| 6 | Cov((-0.1-Sum(close,40)),volume,20)+low | -0.01649 |
| 7 | 0.01Greater(-0.1/Corr(high,close,30),volume) | -0.00823 |
| 8 | Log(|Std((0.05-volume),40)|) | 0.01224 |
| 9 | Greater(-0.01,Log(|Log(|low|)|)) | -0.04616 |
| 10 | Cov(volume,vwap,40) | -0.01412 |

