# OpenReview forum: "Alpha Discovery via Grammar-Guided Learning and Search"
_ICLR.cc/2026/Conference — ICLR 2026 Conference Withdrawn Submission_

### Official Review · Reviewer_A7he · 2025-10-27

**Soundness:** 3
**Presentation:** 2
**Contribution:** 2
**Rating:** 6
**Confidence:** 2

**Summary:**

The paper proposes AlphaCFG, a grammar-guided framework for automated formulaic alpha discovery in quantitative finance. By introducing a context-free grammar with financial semantics, the authors define a structured and interpretable search space for alpha expressions and formulate the discovery task as a tree-structured linguistic Markov decision process. A grammar-aware Monte Carlo Tree Search with Tree-LSTM-based representation learning jointly learns the policy and value functions to efficiently explore this space. Experiments on CSI300 and S&P500 datasets show that AlphaCFG achieves higher information coefficients than existing reinforcement learning and machine learning baselines.

**Strengths:**

1. The paper introduces Context-Free Grammar (CFG) into the automatic discovery of alpha factors for the first time, combining linguistic grammar generation principles with reinforcement learning and Monte Carlo Tree Search (MCTS) to propose a unified framework with both theoretical innovation and practical significance.
2. By defining syntactic and semantic constraints through CFG, the generated alpha factors exhibit clear financial meaning and structural readability, significantly enhancing the interpretability and controllability of the model.
3. The effectiveness of the proposed algorithm is validated on both the Chinese and U.S. stock markets (CSI300 and S&P500), where it consistently outperforms comparative methods in IC, ICIR, and Sharpe Ratio, achieving higher cumulative returns.
4. The overall design is methodologically coherent, with a clear theoretical and implementation path spanning grammar definition, semantic constraints, length control, and reinforcement learning–based search, demonstrating strong research rigor and internal consistency.
5. The supplementary materials provide a well-organized code structure, which facilitates reproducibility of the results.

**Weaknesses:**

1. The problem formulation of alpha discovery could be further improved, as the current description may not provide sufficient clarity for readers who are less familiar with this domain. A more detailed and accessible problem definition would enhance the paper’s readability and impact.
2. The experiments involve numerous hyperparameters, whose tuning is inherently complex and requires extensive search to achieve optimal settings.
3. Both the grammar and reward designs are fixed and manually specified, which limits AlphaCFG’s adaptability to different market conditions and introduces additional human intervention. These components represent extra “dirty work” that undermines the framework’s claimed level of automation.
4. The comparison mainly includes AlphaGen and AlphaQCM as recent baselines, while the others are relatively outdated. The lack of comparison with more recent approaches may underestimate current state-of-the-art performance. In addition, the reference for *gplearn* is missing.

**Questions:**

1. The combination of Grammar-aware MCTS and Tree-LSTM appears computationally expensive when applied to large-scale market features, but the paper does not provide sufficient discussion of the training cost or scalability. Are there any intuitive quantitative comparisons between AlphaCFG and other baselines in terms of efficiency or computational resources?
2. It is not clear whether the defined α-CFG-Sem-k grammar rules can comprehensively represent the common types of alpha constructions in the financial domain, and there may be potential omissions or biases toward certain categories of factors.
3. The reported results show zero variance for XGBoost, which seems implausible and raises concerns about the experimental setup or reporting. Was this due to a single run or to improper averaging.
4. Figure 7 lacks any explanation of experimental reproducibility since the paper does not report the variance or fluctuation of results. Was each experiment executed only once to obtain the presented outcomes.

---

> ### Author Response · Authors · 2025-11-23
> **Response to Reviewer A7he**
>
> **Weakness-1: The problem formulation of alpha discovery could be further improved, as the current description may not provide sufficient clarity for readers who are less familiar with this domain. A more detailed and accessible problem definition would enhance the paper’s readability and impact.**
>
> Thank you for your valuable feedback. We agree that a more detailed and easily understandable problem definition will enhance the readability and influence of the paper. In the revised paper, we have further improved the problem definition section of alpha discovery to increase the clarity and comprehensibility of its expression.
>
> Specifically, we add an example to facilitate understanding what is alpha factor and how it works for non-professional readers in the revised paper. We give an example of an alpha factor (shown in fig5 in Supplementary Material) :Sum(Sub($vwap,1),2d), which sums the values of the current day and yesterday's vwap minus 1. When we want to obtain the yield prediction for Wednesday, we need to calculate the vwap values of Wednesday and Thursday minus 1 and sum them up ((2-1)+(3-1)=3). The resulting value is the predictive value of alpha factor, which is the prediction of the yield for Wednesday. This predicted value will be applied to the subsequent formulation of stock selection strategies.
>
> ---
>
> **Weakness-2: The experiments involve numerous hyperparameters, whose tuning is inherently complex and requires extensive search to achieve optimal settings.**
>
> AlphaCFG does indeed involve a large number of hyperparameters, making tuning difficult and it is hard to achieve the optimal result. However, we conducted experiments using common hyperparameters ( detailed at in Appendix F) and fixed them. The results have been able to exceed the current optimal baseline, and these hyperparameters are provided in the appendix to support the reproduction of the experimental results. Tuning the hyperparameters of the AlphaCFG framework is not our main research content.
>
> Furthermore, although the hyperparameters were not tuned, we tuned the two parameters, the factor pool size and the maximum length of the factor, based on the results of the validation set (detailed at in Appendix H.4.). To a certain extent, this has already led to a relatively better result.
>
> ---
>
> **Weakness-3: Both the grammar and reward designs are fixed and manually specified, which limits AlphaCFG’s adaptability to different market conditions and introduces additional human intervention.  These components represent extra “dirty work” that undermines the framework’s claimed level of automation.**
>
> **Question-2: It is not clear whether the defined α-CFG-Sem-k grammar rules can comprehensively represent the common types of alpha constructions in the financial domain, and there may be potential omissions or biases toward certain categories of factors.**
>
> Please refer to our general response to all reviewers at the beginning. The main purpose of our proposed AlphaCFG is to present a general and extensible formulaic factor search. In our framework:
>
> (1)Grammar is not fixed
> Quantitative researchers can freely add or delete operators, introduce new financial features, adjust semantic constraints, and modify structural rules according to task requirements, that is, naturally integrate domain knowledge into the search process. These modifications only require minor adjustments to the production rules and do not need to change the algorithm structure.
>
> (2)Rewards are not fixed either
> AlphaCFG can use any task-related reward signals such as IC, RankIC, ICIR, maximum drawdown, Sharpe, etc., and can also use multi-objective combinations.
> In the paper, we keep the grammar and rewards fixed merely to ensure a fair comparison with mainstream formulaic factor mining methods such as AlphaGen and AlphaQCM under the same standard settings. The specific operator set and IC rewards used in the paper are just examples in specific tasks and do not cause limitations of the framework itself.
>
> α-CFG-Sem-k combines all the commonly used operators (see Table 5) and features (see Table 3) in the field of stock prediction and is capable of covering general factors. The operators and features are from a well-known work("Generating Synergistic Formulaic Alpha Collections via Reinforcement Learning, Yu et al., SIGKDD 2023").

---

> > ### Author Response · Authors · 2025-11-23
> > **Response to Reviewer A7he**
> >
> > **Weakness-4: The comparison mainly includes AlphaGen and AlphaQCM as recent baselines, while the others are relatively outdated.  The lack of comparison with more recent approaches may underestimate current state-of-the-art performance.  In addition, the reference for gplearn is missing.**
> >
> > We add the reference to the GPlearn method for formulaic factor mining in the revised paper. Regarding the issue of our baseline method being outdated, we clarify: the baseline methods we refer to are all up-to-date, and the misunderstanding was caused by our incorrect references. Our previous citations of baseline methods are incorrect, as they refer to the origins of the machine learning methods rather than their applications in stock prediction. These stock predictions based on machine learning are all relatively new: XGBoost (Wang et al., 2023), LightGBM (Bisdoulis, 2024), LSTM (Bhandariet al., 2022), ALSTM (Qin et al., 2017), TCN (Dai et al., 2022), and Transformer (Mozaffari & Zhang, 2024) . We have updated the citation in the revised paper.
> >
> > ---
> >
> > **Question-1: The combination of Grammar-aware MCTS and Tree-LSTM appears computationally expensive when applied to large-scale market features, but the paper does not provide sufficient discussion of the training cost or scalability. Are there any intuitive quantitative comparisons between AlphaCFG and other baselines in terms of efficiency or computational resources?**
> >
> > All experiments were run on the Inspur NF5280M6 server, with the following configuration:
> > 2× Intel Xeon Silver 4316 (40 cores, 80 threads, 2.3GHz)
> > 251GB of memory
> > 2× NVIDIA GeForce RTX 3090 (24GB ×2)
> >
> > AlphaCFG requires only one RTX 3090 to complete training and search, and each full experiment takes approximately 40 minutes. Our framework has a controllable training and good scalability. Moreover, we compare the usage of computing resources by different methods. We compared AlphaCFG with the latest formulaic factor mining frameworks Alphagen and AlphaQCM, set the maximum length to 10, the factor pool to 10, and recorded the CPU average memory usage and GPU average video memory usage. The results show that Alphagen has an average CPU memory usage of 7.26 GB and an average GPU video memory usage of 0.458 GB; AlphaQCM has an average CPU memory usage of 6.93 GB and an average GPU video memory usage of 1.78 GB; Alphacfg hasan average CPU memory usage of 30.01 GB and an average GPU video memory usage of 11.74 GB. Although AlphaCFG consumes more computing resources compared to the previous method, this does not lead to the limitations of this framework for we can complete the training search using only one RTX 3090.
> >
> > ---
> >
> > **Question-3: The reported results show zero variance for XGBoost, which seems implausible and raises concerns about the experimental setup or reporting. Was this due to a single run or to improper averaging.**
> >
> > To ensure the fairness of all baselines, we conducted experiments using the official default parameters. However, the default parameters of XGBoost do not perform feature sampling (colsample_bytree=1.0) and sample sampling (subsample=1.0). As the default random_state only affects sampling-related modules, the results are consistent under different random number seeds. This phenomenon is caused by the algorithmic properties of XGBoost’s default parameters, rather than experimental flaws.
> >
> > ---
> >
> > **Question-4: Figure 7 lacks any explanation of experimental reproducibility since the paper does not report the variance or fluctuation of results. Was each experiment executed only once to obtain the presented outcomes.**
> >
> > We have repeated all the experiments under five fixed random seeds and reported the mean and variance of each methods’ performance metrics in Table 1. Figure 7 shows the cumulative return curve obtained by selecting the factor with the best IC based on Table 1 from the five repeated experiments of each method and calculating the cumulative return based on the consistent common top-k/drop n trading strategy on qlib (Appendix H.3) after backtesting. We provide all the parameters in Appendix G to support the reproduction of Figure .

---

> > > ### Comment · Reviewer_A7he · 2025-11-28
> > >
> > > Thank you for the detailed responses. Your clarifications helped me better understand several important aspects of AlphaCFG, including its cost considerations and experimental details. Based on this improved understanding, I will maintain my positive overall assessment and increase my confidence score accordingly.

---

> > > > ### Author Response · Authors · 2025-11-28
> > > >
> > > > Thank you for maintaining your positive assessment and increasing your confidence.  We are truly grateful that​ your clarifications helped us improve the presentation of AlphaCFG’s cost considerations and experimental details in the paper.  We remain available to clarify other points if necessary.

---

### Official Review · Reviewer_iYe2 · 2025-10-31

**Soundness:** 3
**Presentation:** 3
**Contribution:** 2
**Rating:** 2
**Confidence:** 4

**Summary:**

This paper introduces AlphaCFG, a grammar-constrained framework for discovering alpha factors in quantitative finance. The method defines a domain-aware context-free grammar (α-CFG-Sem-k) to generate syntactically and semantically valid formulaic factors within a bounded search space. Alpha generation is formulated as a Tree-Structured Linguistic MDP (TSL-MDP), where leaf rewards correspond to the factor’s Information Coefficient (IC). The MDP is solved using a grammar-aware Monte Carlo Tree Search (MCTS) guided by Tree-LSTM–based policy and value heads that evaluate partial expressions. The authors further include a similarity regularizer based on the largest common subtree to promote diversity among discovered alphas. Experiments on CSI300 and S&P500 show that AlphaCFG achieves higher IC and backtest profitability than several baselines (AlphaGen, AlphaQCM, GPlearn, XGBoost), and it can refine existing factors such as GTJA191 and Alpha101.

**Strengths:**

- The use of a context-free grammar (CFG) grounded in financial semantics effectively constrains the search space, ensuring valid and interpretable formulas while reducing redundancy compared to unrestricted symbolic approaches. Modeling formula generation as a Tree-Structured Linguistic MDP (TSL-MDP) offers a clean theoretical lens for combining RL with symbolic expression search, justifying the use of MCTS and policy/value learning.
- Using Tree-LSTM–based policy and value estimators enables partial expression evaluation and improves search efficiency. The subtree-based similarity regularization is also a thoughtful mechanism to enhance formula diversity.
- The experiments benchmark AlphaCFG against a broad set of symbolic, ML, and RL-based methods across two major indices, with multiple metrics reported (IC, Sharpe, MaxDD). The demonstration on refining existing alphas adds practical relevance.

**Weaknesses:**

- Semantic equivalence pruning lacks rigor and scalability analysis. The paper uses subtree-based similarity for pruning equivalent or redundant expressions but does not analyze its computational complexity or describe practical optimizations (e.g., hashing or canonicalization). Given the large candidate pool, this omission is a serious limitation.
- Restrictive grammar and operator set. The experiments use only six primitive features and a small operator vocabulary, which raises concerns about overfitting and limited expressivity. The paper does not demonstrate scalability when introducing richer operators or additional financial features.
- Reward design misaligned with trading objectives. Optimizing solely for IC ignores turnover and transaction costs. Although backtests show strong returns, these metrics are not optimized directly, leaving doubts about robustness under realistic frictions.
- Weak ablations and architectural justification. The choice of Tree-LSTM is not well motivated relative to other plausible encoders (e.g., GNNs or transformers). Ablations on encoder type, embedding size, and similarity penalty strength are missing, weakening the causal claims about representation quality.

**Questions:**

- How is the semantic similarity function computed efficiently at scale? What is its time complexity, and are approximate or hashing-based methods used?
- What are the compute resources (GPUs, CPUs, wallclock time) and total number of IC evaluations used for AlphaCFG versus AlphaGen/AlphaQCM?
- How sensitive are results to the choice of operators and constants? Would adding common financial primitives (e.g., liquidity or volatility) change outcomes?
- Were results averaged across multiple seeds, and what is the variance? Please report statistical tests versus the strongest baselines.
- In the factor-refinement experiments, how much modification occurs to the original factor, and how do you ensure these changes are not trivial rewrites that merely inflate IC?


I would consider raising my score if the authors can adequately address these questions.

---

> ### Author Response · Authors · 2025-11-23
> **Response to Reviewer iYe2**
>
> **Weakness-1: Semantic equivalence pruning lacks rigor and scalability analysis. The paper uses subtree-based similarity for pruning equivalent or redundant expressions but does not analyze its computational complexity or describe practical optimizations (e.g., hashing or canonicalization). Given the large candidate pool, this omission is a serious limitation.**
>
> Thank you for raising questions on the computational complexity.  We clairify that, although the space of all possible alpha factors generated by α-CFGs is very large (it is essentially infinite if one does not constrain the length of the factor), we have reduced the search space by the following key operations.
>
> (1)Complexity reduction in the alpha factor modeling stage: we have provided an adjustable alpha factor length K, within which the generated alpha factor are not only valid, but also their space is much shrinked. This is also reasonable, since we aim to provide human-interpreteable alpha factors, a very large length is not under consideration.
>
> (2)Complexity reduction in the alpha factor representation stage: we have applied Tree-LSTM to ENCODE the abstract syntax tree (i.e., each node on the MCTS search space) to obtain a vector with a fixed length. This has exactly the same effect of your suggested optimizations such as hashing or canonicalization, which significantly compress the huge search tree. More importantly, Tree-LSTM is a machine learning approach, it can provide performance oriented optimization. Please kindly refer to Section 4.4 and Appendix E for the details.
>
> (3)Complexity reduction in the tree-search stage: After the Tree-LSTM representation learning, during our search scheme by MCTS+RL, the search space is already much reduced by the above two operations, and is search tree with much-controlled width and depth.  Importantly, the similarity we calculate during the MCTS is NOT based on ALL the factors, but on a fixed number of factors in the “factor pool”. Therefore, the computational complexity is constant.
>
> We have added a detailed complexity analysis in Appendix D,please see the updated paper, and you can also refer to our response to reviewer-1 about the complexity.
> With the above four optimization steps and the controlled complexity, the alpha factor mining task can be easily achieved by a PC with NVIDIA GeForce RTX 3090 (please refer to our response to reviewer-1 on the Computational resources). We will follow your comment and clarify the above issues in the updated version.
>
> ---
>
> **Weakness-2: Restrictive grammar and operator set. The experiments use only six primitive features and a small operator vocabulary, which raises concerns about overfitting and limited expressivity. The paper does not demonstrate scalability when introducing richer operators or additional financial features.**
>
> **Weakness-3: Reward design misaligned with trading objectives. Optimizing solely for IC ignores turnover and transaction costs. Although backtests show strong returns, these metrics are not optimized directly, leaving doubts about robustness under realistic frictions.**
>
> For these two concerns, please refer to our general response in the beginning. The reviewers seem concerned that our framework can only operate with fixed grammar and rewards, but this is not true. AlphaCFG is a flexible general framework that allows for adjustments to both operators and rewards. The grammar and reward settings in this paper are kept simple and basic to let the readers focus on the framework itself but not on specific finance task. The simple setting of these two also aim to give a fair comparison with existing methods like AlphaGen and AlphaQCM. According to your comment, we will carefully revise the writing of the paper and emphasize this clearly.
>
> Regarding operator extensions, quantitative finance researchers can easily add or remove operators, introduce new financial features, and adjust semantic constraints or structural rules to suit task requirements. These modifications only require minor changes to the production rules, without affecting the core algorithm.
>
> For reward designs, AlphaCFG supports various task-specific reward signals, such as IC, RankIC, ICIR, maximum drawdown, Sharpe ratio, and multi-objective combinations. The fixed grammar and rewards in the paper are just for comparison purposes and do not limit the framework's flexibility.

---

> ### Author Response · Authors · 2025-11-23
> **Response to Reviewer iYe2**
>
> **Weakness-4: Weak ablations and architectural justification. The choice of Tree-LSTM is not well motivated relative to other plausible encoders (e.g., GNNs or transformers). Ablations on encoder type, embedding size, and similarity penalty strength are missing, weakening the causal claims about representation quality.**
>
> In the revised version of our paper, we add ablation studies on the network architecture (Section 5).
>
> We conducted comparative experiments under different network architectures (Transformer, LSTM, CNN) while keeping other conditions constant. With a pool size of 10 and max length 5, fig3 in Supplementary Material shows training IC across epochs. Results demonstrate the effectiveness and superiority of syntax representation learning. Tree-LSTM not only extracts the semantic information of expressions but also reduces tree-structural redundancy caused by similar Abstract Syntax Trees representing the alpha factors.
>
> Besides this added experiment, we also added other ablation studies about factor length K, please refer to our response to reviewer-eWNQ.
>
> ---
>
> **Question-1: How is the semantic similarity function computed efficiently at scale? What is its time complexity, and are approximate or hashing-based methods used?**
>
> Please see our answer to your comment-1 above.
>
> ---
>
> **Question-2: What are the compute resources (GPUs, CPUs, wallclock time) and total number of IC evaluations used for AlphaCFG versus AlphaGen/AlphaQCM?**
>
> All experiments were run on the Inspur NF5280M6 server, with the following configuration:
> 2× Intel Xeon Silver 4316 (40 cores, 80 threads, 2.3GHz)
> 251GB of memory
> 2× NVIDIA GeForce RTX 3090 (24GB ×2)
> AlphaCFG requires only one RTX 3090 to complete training and search, and each full experiment takes approximately 40 minutes. Our framework has a controllable training and good scalability.  AlphaCFG generates 100 factor expressions for 100 IC evaluations in each epoch.
>
> Moreover, we compare the usage of computing resources by different methods. We compared AlphaCFG with the latest formulaic factor mining frameworks Alphagen and AlphaQCM, set the maximum length to 10, the factor pool to 10, and recorded the CPU average memory usage and GPU average video memory usage. The results show that Alphagen has an average CPU memory usage of 7.26 GB and an average GPU video memory usage of 0.458 GB; AlphaQCM has an average CPU memory usage of 6.93 GB and an average GPU video memory usage of 1.78 GB; Alphacfg hasan average CPU memory usage of 30.01 GB and an average GPU video memory usage of 11.74 GB. Although AlphaCFG consumes more computing resources compared to the previous method, this does not lead to the limitations of this framework for we can complete the training search using only one RTX 3090.
>
> ---
>
> **Question-3: How sensitive are results to the choice of operators and constants? Would adding common financial primitives (e.g., liquidity or volatility) change outcomes?**
>
> To answer this question,
>
> (1)We first present a theoretical analysis: Adding new operators and features to AlphaCFG essentially refers to the combination of basic operators and basic features . If the combined operators or features are very common in stock prediction and have good predictive capabilities, then adding such operators will make the results better. Essentially this is shortening the path on the MCTS tree of high-quality factors. Here, basic features refer to those directly obtained from stock data without any further computation, while basic operators denote primitive operations in mathematics that are not defined based on other operators.
>
> (2)We conduct an experiment to compare the α-CFG after removing all the operators that can be obtained from the combination of basic operators with the initial version and get the figure below. The results (shown in fig4 in Supplementary Material) are consistent with our previous analysis. When combining the basic operators to obtain the common operators in stock prediction, it makes the search more efficient and it is easier to search for factors with high IC.

---

> ### Author Response · Authors · 2025-11-23
> **Response to Reviewer iYe2**
>
> **Question-4: Were results averaged across multiple seeds, and what is the variance? Please report statistical tests versus the strongest baselines.**
>
> The results in our Table1 are the results of multiple experiments with five random number seeds. The mean is in front of the parentheses, and the standard deviation is in the parentheses (XGBoost method has no randomness because the baseline uses default parameters). Compared with the current strongest baseline AlphaQCM, our AlphaCFG is better in both CSI 300 and S&P 500. On CSI 300, the Rank IC is 0.0865 (±0.0060), the IC is 0.0577 (±0.0029), the Rank ICIR is 0.6036 (±0.0537), the ICIR is 0.4505 (±0.0249), the Sharpe ratio is 0.6459 (±0.0612), the Maximum Drawdown is -0.2963 (±0.0289).  On S&P 500, the Rank IC is 0.0354 (±0.0026), the IC is 0.04573 (±0.0034), the Rank ICIR is 0.2958 (±0.0154), the ICIR Is 0.4099 (±0.0230), the Sharpe ratio was 0.8473 (±0.0483), the Maximum Drawdown Is -0.1942 (±0.0126). Our indicator testing has better mean values, smaller standard deviations, which means our framework is more efficient and stable.
>
> ---
>
> **Question-5: In the factor-refinement experiments, how much modification occurs to the original factor, and how do you ensure these changes are not trivial rewrites that merely inflate IC?**
>
> In the factor-refinement experiments, Factor modification reserves partial structure of the classic well-known alpha factors, it mask the symbols from right to left to obtain a refined alpha factor which has length of half the length of the alpha factor and then use the incomplete expression to continue the search with IC as the optimization objectove. Here, it can be understood that we retain the nodes whose length in the search tree path corresponding to the classic factors does not exceed half, that is, only the path structure of the first half is intercepted. Subsequently, taking the very end node of this truncated path as the new root node, we carry out a new search.
> The core idea is : searching for a high-quality factor is equivalent to finding a path to a leaf node with high IC value in the search tree. The validity of the classic factor (i.e., a path that has been verified to be effective) has been confirmed. Therefore, even if this factor gradually becomes ineffective in the current market, we can still start the search from the first few layers of this path again. This strategy is reasonable. This approach can essentially reduce the scale of the search space. Instead of blindly searching the whole tree, it is better to search based on the "prefix" of an existing effective structure, that is, to search a smaller subtree, and the search efficiency is also higher.
>
> Regarding how to ensure that the factors modified in the above-mentioned way are not merely to enhance IC but to make meaningful modifications, this is not our main research approach. However, we offer some idea: evaluating and filtering the modified factors based on multiple indicators such as ICIR, maximum drawdown or manual screening. Essentially, this is not different from directly searching for formulaic factors throughout the entire search space. However, this perspective based on the modification of classical factors provides a new idea for factor mining tasks from the perspective of reducing the search space.

---

### Official Review · Reviewer_eWNQ · 2025-11-01

**Soundness:** 2
**Presentation:** 2
**Contribution:** 3
**Rating:** 6
**Confidence:** 4

**Summary:**

This paper introduces AlphaCFG, a framework for automated discovery of interpretable, high-performing formulaic alpha factors using a context-free grammar for syntactic/semantic constraints, coupled with a reinforcement learning-guided MCTS. AlphaCFG explicitly formalizes the syntax and semantics of valid financial expressions, bounds search complexity via grammar and expression length, and employs Tree-LSTM–based neural networks as value and policy estimators within MCTS. The approach is empirically evaluated on China (CSI 300) and US (S&P 500) equities, further showcases the framework’s ability to refine classic domain factors.

**Strengths:**

1.	Principled, Structured Search via Grammar: By leveraging domain-specific rules—such as operator arity, finance logic, and context-dependent operand constraints, the paper addresses a central limitation in prior factor mining efforts: managing the combinatorial explosion and redundancy of candidate spaces.
2.	Integration of Reinforcement Learning and Neural MCTS: The formulation of alpha search as a Tree-Structured Linguistic MDP and the embedding of neural MCTS with Tree-LSTM–based networks offer a methodologically coherent and scalable solution for symbolic expression discovery.
3.	Empirical Rigor Across Multiple Markets and Metrics: The paper provides comprehensive head-to-head comparisons with existing baselines. The results, across correlation metrics (IC, RankIC) and trading metrics, consistently favor the proposed approach, and the ablation studies convincingly validate the role of syntax, semantics, and length controls.
4.	Interpretability and Factor Refinement: The capacity to refine existing domain factors shows the practical relevance and transferability of the framework, highlighting the potential for human-in-the-loop discovery and sector adoption.

**Weaknesses:**

1.	The paper lacks a quantitative report (e.g., a 1% syntax error rate) to conclusively prove that all generated expressions are syntactically valid
2.	The paper extends the grammar to α-CFG-Sem (Definition 3) by incorporating domain-specific semantic constraints. The designed constraints are heuristically sound and based on reasonable financial logic. However, the connection between these specific constraints and the formal property of "interpretability" is not rigorously proven.
3.	Although Sharpe and Max Drawdown are computed, there is no investigation into the impact of these real-world trading constraints or their explicit inclusion within the search objective/loss.
4.	Scope and Generalization: The framework’s evaluation is limited to two well-established benchmarks (CSI 300, S&P 500). There is no evidence or discussion concerning the transferability to other financial datasets, time periods, asset classes, or tasks outside formulaic alpha mining—whereas some prior works position their language discovery techniques as generally applicable beyond their primary domain.
5.	Figure 6 demonstrates faster convergence for CFG methods compared to RPN, suggesting improved sample efficiency. However, the paper lacks direct evidence of computational efficiencyNo comparison of wall-clock time or CPU/GPU hours against baselines, No analysis of memory usage. No theoretical complexity analysis relating to the size of the grammar-defined search space.

**Questions:**

1.	Could the authors provide examples of trading strategies (alphas) with clear financial intuition, along with an analysis of their performance during specific market regimes?
2.	Could the authors clarify (with ablation or added experiments) the practical impact of the maximum length and semantic constraints? Are some informative or profitable alphas systematically excluded by the grammar?
3.	How robust is the AlphaCFG approach to changes in the underlying data regime (e.g., bull vs. bear periods, out-of-distribution price actions)? Can the authors provide empirical results/data splits that systematically probe regime sensitivity?

---

> ### Author Response · Authors · 2025-11-23
> **Response to Reviewer-eWNQ**
>
> **Weakness-1: The paper lacks a quantitative report (e.g., a 1% syntax error rate) to conclusively prove that all generated expressions are syntactically valid**
>
> In formal language theory, “syntactically valid” means that an expression (a final alpha factor) can be derived from the start symbol of a formal grammar by applying the grammar’s production rules step by step and must contain only terminal symbols.
> In our setting, all alpha expressions in the search space are syntactically valid, as they are generated exclusively by the α-CFG grammars (Definitions 2 and 3 and Algorithm 2 with constraint K).
>
> Therefore, we formally characterize the space of α-CFG alpha expressions, all of which are inherently syntactically valid, eliminating any risk of “syntax errors.” The first half of the paper develops a linguistic-theoretic framework (i.e., the α-CFGs) that guarantees all possible expressions in the MCTS search space (within depth K) is syntactically valid. In the second half of our paper, on this vast MCTS tree, we mine the high-performance alpha factors. No matter how we optimize the MCTS, there is no possibility of a “syntax error” in the formal sense, because there is no way to produce a string outside the CFG language space. In practical applications, users only need to consider the exact length of factor to stop expansion and ensure that the final expression contains no non-terminal symbols.
>
> ---
>
> **Weakness-2: The paper extends the grammar to α-CFG-Sem (Definition 3) by incorporating domain-specific semantic constraints. The designed constraints are heuristically sound and based on reasonable financial logic. However, the connection between these specific constraints and the formal property of "interpretability" is not rigorously proven.**
>
> The interpretability of our framework means that our obtained high-performance formulaic alpha factors (1) have explicit formula form and (2) are syntactically valid guaranteed by formal language α-CFGs (3) have human-acceptable length K. These enable financial practitioners to understand the meaning of these factors.
>
> Syntactically valid is a basic constraint in AlphaCFG that every generated alpha factor should satisfy. However, lacking semantic constraints, some expressions (e.g., nesting multiple operators on constants and performing correlation operations on constants) do not have practical financial significance and thus are unexplainable.
>
> To extend our basic syntactically valid α-CFG to semmatically valid factors, we introduce domain-specific semantic constraints intoα-CFG-Sem-K (Definition 4), we ensure that all generated expressions are both syntactically valid and semantically reasonable. These constraints effectively filter out expressions that are meaningless in finance, thereby enhancing the interpretability of the factors.
> We will state this more clearly in the revised version.
>
> ---
>
> **Weakness-3： Although Sharpe and Max Drawdown are computed, there is no investigation into the impact of these real-world trading constraints or their explicit inclusion within the search objective/loss.**
>
> **Weakness-4:Scope and Generalization: The framework’s evaluation is limited to two well-established benchmarks (CSI 300, S&P 500). There is no evidence or discussion concerning the transferability to other financial datasets, time periods, asset classes, or tasks outside formulaic alpha mining—whereas some prior works position their language discovery techniques as generally applicable beyond their primary domain.**
>
> **Question-3:How robust is the AlphaCFG approach to changes in the underlying data regime (e.g., bull vs. bear periods, out-of-distribution price actions)? Can the authors provide empirical results/data splits that systematically probe regime sensitivity?**
>
> Please refer to our general response to all reviewers at the beginning. The main objective of this paper is to propose an automated learning + search framework for linguistic-theory-based alpha factor discovery. In other words, AlphaCFG is a general and flexible framework and which supports various domains and different experiment settings, it is not simply a specific trading strategy.
>
> Our experiments on the CSI 300 and S&P 500 are provided as illustrative case studies, primarily to demonstrate to the readers the effectiveness of AlphaCFG framework. This does not aim at providing a specific trading strategy. Users can also apply AlphaCFG to other domains.

---

> ### Author Response · Authors · 2025-11-23
> **Response to Reviewer-eWNQ**
>
> **Weakness-5：Lack of theoretical complexity analysis and concerns of computational efficiency**
>
> (1) Theoretical complexity. In response to your comment, we have added a detailed complexity and runtime analysis. Specifically, we supplement the discussion of theoretical complexity in Appendix D of the revised paper.
> In general, discovering any formulaic expression is a combinatorial problem, and the computation time grows exponentially with the length of the target formula. It implies that, without constraints, the search tree for MCTS grows exponentially in the length of the alpha factor. Please note that this is the common challenge to all researches of formulaic alpha factors.
> However, our linguistic framework, α-CFG-Sem restricts the alpha factors to length at most (K), while still enforces syntactic validity and. As a result, under AlphaCFG, the computational time becomes constant with respect to (K). The exact number of basic computation is provided in the updated manuscript based on a recursive analysis (Appendix D), and we additionally include a figure (fig1 in Supplementary Material) illustrating how the computation time increases.
>
> (2) Computational resources:  All experiments were run on the Inspur NF5280M6 server, with the following configuration:
> 2× Intel Xeon Silver 4316 (40 cores, 80 threads, 2.3GHz)
> 251GB of memory
> 2× NVIDIA GeForce RTX 3090 (24GB ×2)
> AlphaCFG requires only one RTX 3090 to complete training and search, and each full experiment takes approximately 40 minutes. Our framework has a controllable training and good scalability. Moreover, we compare the usage of computing resources by different methods. We compared AlphaCFG with the latest formulaic factor mining frameworks Alphagen and AlphaQCM, set the maximum length to 10, the factor pool to 10, and recorded the CPU average memory usage and GPU average video memory usage. The results show that Alphagen has an average CPU memory usage of 7.26 GB and an average GPU video memory usage of 0.458 GB; AlphaQCM has an average CPU memory usage of 6.93 GB and an average GPU video memory usage of 1.78 GB; Alphacfg hasan average CPU memory usage of 30.01 GB and an average GPU video memory usage of 11.74 GB. Although AlphaCFG consumes more computing resources compared to the previous method, this does not lead to the limitations of this framework for we can complete the training search using only one RTX 3090.
>
> ---
>
> **Question1: Could the authors provide examples of trading strategies (alphas) with clear financial intuition, along with an analysis of their performance during specific market regimes**
>
> For your concern, we provide an example of an alpha factor mined by our framework, together with a detailed interpretation, in Appendix H.5.
> Intuitively, a factor obtained by our AlphaCFG with high performance is
> **Log(|Std((0.05 − volume), 40)|)**
> measures the volatility of inverse trading volume over a 40-day window. This factor captures the temporal variability of illiquidity, which may signal periods of market stress or large price-impact events.
> Another example,
> **Cov(volume, vwap, 40)**,
> reflects the co-movement between trading volume and the volume-weighted average price. A large covariance suggests strong directional consensus, which may correspond to persistent momentum or potential price reversals.
>
> We also attach the top 10 alpha factors discovered by AlphaCFG. Since they are relatively short and strictly follow the CFG grammar, each of them can be interpreted in a straightforward manner in standard financial language.
>
> | #  | Alpha Expression | Weight   |
> |----|------------------|----------|
> | 1  | Mean(Corr(Sum(open,40),(high-volume),20),20) | -0.00889 |
> | 2  | volume | -0.01278 |
> | 3  | Std(close,40) | 0.01778 |
> | 4  | Pow(Med(Cov(high,low,30),30),0.1) | 0.01411 |
> | 5  | Delta(Log(\|Min(high,30)/0.01\|),30) | -0.01649 |
> | 6  | Cov((-0.1-Sum(close,40)),volume,20)+low | -0.01649 |
> | 7  | 0.01Greater(-0.1/Corr(high,close,30),volume) | -0.00823 |
> | 8  | Log(\|Std((0.05-volume),40)\|) | 0.01224 |
> | 9  | Greater(-0.01,Log(\|Log(\|low\|)\|)) | -0.04616 |
> | 10 | Cov(volume,vwap,40) | -0.01412 |
>
> **Table:** Top 10 Ranked Alphas and Their Weights
>
> We didn’t do investigation for AlphaCFG’s performance specific market regime, please kindly refer the our general response to all reviewers in the beginning, where we explained that this is because we aim to provide a general framework and this framework is adaptable to different specific finance tasks. Users who are interested in different states of markets can quickly adapt AlphaCFG to their market regime settings.

---

> ### Author Response · Authors · 2025-11-23
> **Response to Reviewer-eWNQ**
>
> **Question2: Could the authors clarify (with ablation or added experiments) the practical impact of the maximum length and semantic constraints? Are some informative or profitable alphas systematically excluded by the grammar?**
>
> To address your question, we include additional experiments in Appendix H.4, where we compare different levels of α-CFG grammars against the RPN baseline. Specifically, we conduct experiments on the validation set along two dimensions: the **maximum length of individual factors** (Max Length) and the **factor pool size** (Pool Size), with results reported in fig2 in Supplementary Material.
>
> We first fix the maximum length of individual factors and evaluate the validation IC for different pool sizes {1, 5, 10, 20, 30} to select the optimal pool size. After determining the optimal pool size under CFG-SSL, we fix this pool size and then vary the maximum length of individual factors {5, 10, 15, 20, 25} to identify the best configuration. In this process, the variation along the x-axis reveals the influence of the maximum expression length on IC, while the different curves reflect the impact of semantic constraints.
>
> These results directly demonstrate the effects of maximum length and semantic constraints on factor search performance. The constraints we design only filter out syntactically or semantically invalid expressions and do not exclude potentially effective alpha factors.

---

### Author Response · Authors · 2025-11-23
**General response to all reviewers**

We are deeply grateful to the reviewers. We have addressed each concern in the following point-by-point responses.  We also carefully revised the manuscript. Please see the red colored contents in the attached new version of the paper.

We would like to clarify that the primary contribution of this paper is NOT ONLY a high-performing trading scheme, but a **general “linguistic theory + learning + search” framework for generating formulaic alpha factors (AlphaCFG) for various tasks in quantitative finance**. Trading is an important application of this framework, but not the sole target. Consistent with this objective, we attach only a simple, basic top-k/drop-n trading strategy (Appendix H.3). In the experiment, you can see that even attached with this basic trading scheme, AlphaCFG still obtains competitive trading performance.

More importantly, AlphaCFG is a flexible grammar framework, not restricted to the specific operators in Definitions 2 and 3. These operators are only illustrative, chosen to make the implementation of AlphaCFG easier to understand (also, they are following a well-known work "Generating Synergistic Formulaic Alpha Collections via Reinforcement Learning, Yu et al., SIGKDD 2023"). Users can plug in their own domain-specific operators. For example, operators from risk modelling, portfolio construction, or asset pricing can be used in Definitions 2 and 3 without changing the AlphaCFG framework itself.

For the same reason, we use IC as a convenient, standard evaluation metric, but users can readily substitute alternative criteria and modify the loss functions in the MCTS component to match their own objectives. In all these cases, the framework remains the same; only the domain knowledge and evaluation goals change.

---

### Note · Authors · 2026-01-14

I have read and agree with the venue's withdrawal policy on behalf of myself and my co-authors.